# Developmental Federated Tuning: A Cognitive-Inspired Paradigm for Efficient LLM Adaptation

**Yebo Wu**[1,*], **Jingguang Li**[2,*], **Zhijiang Guo**[3,4,†] , **Li Li**[1,†]
[1]University of Macau  [2]KAIST  [3]HKUST  [4]HKUST (Guangzhou)
{yc37926, llili}@um.edu.mo
jingguangli2001@gmail.com, zhijiangguo@hkust-gz.edu.cn

## Abstract

Federated fine-tuning enables Large Language Models (LLMs) to adapt to downstream tasks while preserving data privacy, but its resource-intensive nature severely limits deployment on edge devices. In this paper, we introduce Developmental Federated Tuning (DevFT), a resource-efficient approach inspired by cognitive development that progressively builds a powerful LLM from a compact foundation. DevFT decomposes the fine-tuning process into developmental stages, each optimizing a submodel with increasing parameter capacity. Knowledge acquired in earlier stages is transferred to subsequent submodels, providing optimized initialization parameters that prevent convergence to local minima and accelerate training. This paradigm mirrors human learning, gradually constructing a comprehensive knowledge structure while refining existing skills. To efficiently build stage-specific submodels, DevFT introduces deconfliction-guided layer grouping and differential-based layer fusion to distill essential information and construct representative layers. Evaluations across multiple benchmarks demonstrate that DevFT significantly outperforms state-of-the-art methods, achieving up to **4.59×** faster convergence, **10.67×** reduction in communication overhead, and **9.07%** average performance improvement, while maintaining compatibility with existing federated fine-tuning approaches.

## 1 Introduction

Large Language Models (LLMs) exhibit exceptional capabilities across diverse domains (Yuan et al., 2024; Xu et al., 2025; 2024b). While fine-tuning effectively adapts these models to specific tasks (Han et al., 2024), it demands substantial task-specific data. This data often resides privately on edge devices, making centralized collection impractical. Federated fine-tuning (Zhang et al., 2024a) offers a privacy-preserving alternative for collaborative adaptation. Nevertheless, deploying massive LLMs for federated fine-tuning on resource-limited edge devices is challenging due to hardware and communication constraints (Wu et al., 2025a; Tam et al., 2024; Tian et al., 2024a; 2022).

To address these challenges, researchers have proposed various parameter-efficient federated fine-tuning approaches (Wu et al., 2025d), with LoRA-based methods garnering significant attention due to their efficiency and flexibility (Guo et al., 2025; Tian et al., 2024b). However, existing LoRA-based methods typically fine-tune LLMs end-to-end, which remains computationally prohibitive for edge devices compared to small language models such as BERT (Devlin et al., 2019). Figure 1 quantifies this gap by comparing the computational cost of a single fine-tuning step across LLaMA (Touvron et al., 2023) variants and BERT. Even the relatively compact TinyLLaMA (Zhang et al.,

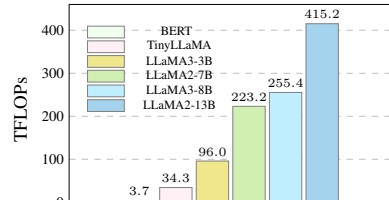

Figure 1: Computational overhead for one-step fine-tuning of different language models using LoRA.

---

*Equal Contribution.    †Corresponding Authors.

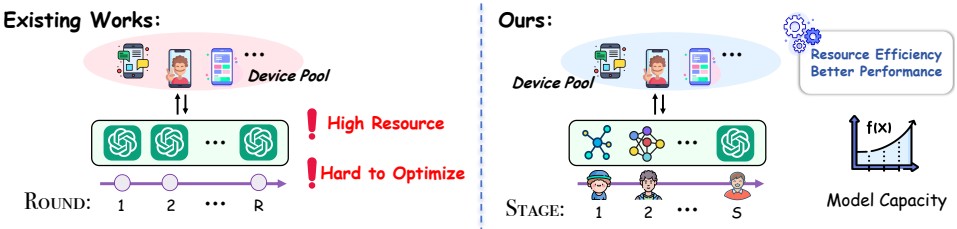

Figure 2: Workflow comparison between existing works and DEVFT.

2024b) demands $9.3\times$ more FLOPs than BERT, while LLaMA2-13B (Touvron et al., 2023) requires an overwhelming 415.2 TFLOPs, which is $112.2\times$ that of BERT. Such substantial computational requirements fundamentally challenge the practical deployment of federated fine-tuning on resource-constrained devices, even with current parameter-efficient techniques.

Inspired by human cognitive development (Bengio et al., 2009; Sweller, 2008; McArdle & Woodcock, 2014), where learning unfolds progressively rather than instantaneously, we propose **Developmental Federated Tuning (DEVFT)**, a resource-efficient federated fine-tuning approach that alleviates computational burdens by gradually cultivating a capable LLM from a compact foundation. As illustrated in Figure 2, instead of updating the full LLM throughout the entire federated fine-tuning process, DEVFT decomposes learning into a sequence of developmental stages. Specifically, the learning journey begins with a compact submodel (*analogous to a child*), and upon mastering stage-specific competencies, we strategically expand the submodel capacity (*mimicking human growth*), while transferring the acquired knowledge to initialize the submodel of the next stage. This growth-and-transfer process repeats until the model reaches its target capacity (*analogous to an adult*).

This developmental paradigm, starting with compact models, offers several inherent advantages. Smaller models generally exhibit smoother loss landscapes, reducing the risk of convergence to poor local minima. Moreover, the knowledge distilled during early stages serves as a well-informed initialization for larger architectures, enhancing performance in subsequent stages. Compared to end-to-end LLM fine-tuning, DEVFT 's progressive capacity growth substantially accelerates federated fine-tuning while lowering both computation and communication overheads. However, a critical challenge lies in: *How to architect stage-specific submodels to ensure effective knowledge transfer across consecutive stages while optimizing overall performance?*

To address this challenge, DEVFT introduces two novel techniques. The deconfliction-guided layer grouping mechanism initially clusters layers based on parameter similarity, thereby grouping layers with minimal parameter conflicts together. Subsequently, the differential-based layer fusion strategy strategically distills and integrates the distinctive semantic information of each layer through arithmetic operations, yielding a representative layer for each group that encapsulates the group's collective knowledge and core functionality. These representative layers are then concatenated sequentially to construct the stage-specific submodel for federated fine-tuning. Due to the functional homogeneity within groups, layers can directly inherit knowledge from their corresponding representative layers, thereby facilitating seamless knowledge transfer across developmental stages.

In order to empirically validate the effectiveness of DEVFT and its advantages, we conduct extensive experiments on multiple benchmarks. DEVFT significantly outperforms state-of-the-art methods, achieving up to $4.59\times$ faster convergence, $10.67\times$ reduction in communication overhead, and $9.07\%$ average performance improvement, while maintaining compatibility with existing approaches.

## 2 BACKGROUND AND MOTIVATION

### 2.1 EXISTING PARAMETER-EFFICIENT FEDERATED FINE-TUNING

Parameter-efficient federated fine-tuning presents a compelling strategy to mitigate resource demands in distributed learning by freezing most pre-trained model parameters and updating only a small, task-specific subset (Wu et al., 2025d). These methods generally fall into the following categories. Prompt-based techniques (Guo et al., 2023; Yang et al., 2023; Su et al., 2024) utilize carefully designed soft prompts to guide model behavior without altering the pre-trained weights. Adapter-

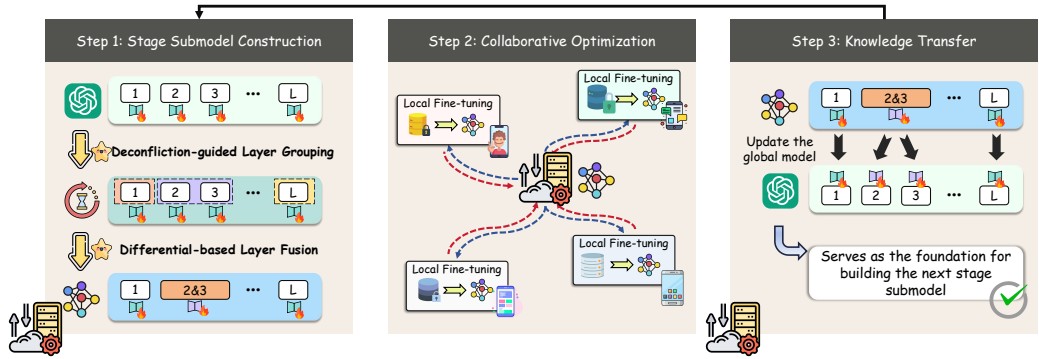

Figure 3: Overview of DEVFT: The server first constructs the stage-specific submodel (step ①), followed by collaborative optimization across edge devices (step ②). After each stage, the acquired knowledge is employed to update the global model, which serves as the foundation for building the subsequent stage submodel (step ③).

based methods (Cai et al., 2023; Liu et al., 2023; Li et al., 2022) insert lightweight adapter layers into the network, allowing for task adaptation with minimal modifications. Notably, LoRA-based approaches have garnered significant interest due to their effectiveness (Guo et al., 2025).

LoRA-based methods (Wang et al., 2024; Sun et al., 2024; Wu et al., 2025b) introduce low-rank adaptations to weight updates, effectively preserving the expressiveness of the original model while significantly reducing the number of trainable parameters. To accommodate heterogeneous resources, approaches like HETLoRA (Cho et al., 2024) and FlexLoRA (Bai et al., 2024) assign varying LoRA ranks to different devices. Fed-pilot (Zhang et al., 2024c) and Fed-HeLLo (Zhang et al., 2025) optimize LoRA allocation through layer contribution quantification and resource-aware importance scoring. Moreover, FwdLLM (Xu et al., 2023) and FedKSeed (Qin et al., 2023) employ zeroth-order optimization to mitigate resource consumption. Furthermore, FeDeRA (Yan et al., 2024) addresses data heterogeneity by initializing LoRA via singular value decomposition on pre-trained parameters. While these methods have shown promise in their respective domains, they often do not fully tackle the fundamental issue of the substantial computational requirements imposed by end-to-end LLM fine-tuning. This persistent challenge motivates our proposed approach.

## 2.2 MOTIVATION FOR DEVELOPMENTAL FEDERATED TUNING

To address the persistent challenge of substantial computational burdens, we propose a different paradigm. Unlike existing works that update the LLM in an end-to-end manner, which can be resource-intensive, our approach progressively builds a capable model from a compact foundation. Drawing inspiration from human cognitive development (Bengio et al., 2009; Sweller, 2008), where learning progresses incrementally rather than instantaneously, we aim to mitigate these computational burdens by progressively cultivating a more capable LLM from a compact foundation.

Specifically, we decompose the fine-tuning process into $S$ stages, each with multiple rounds, mimicking different periods in human learning. The submodel capacity (i.e., the number of layers) at each stage is denoted as $\{L_1, L_2, \ldots, L_S\}$, forming a strictly monotonically increasing sequence where $L_{s_1} < L_{s_2}$ for any $s_1 < s_2$. The final stage capacity $L_S$ equals $L$, encompassing all layers of the LLM. Additionally, the knowledge acquired in each stage seamlessly transfers to the submodel of the subsequent stage, providing optimized initialization parameters. Compared to end-to-end fine-tuning, this developmental paradigm significantly reduces resource overhead for edge devices while achieving superior performance through a smoother optimization trajectory. In this way, DEVFT enables participating devices to efficiently fine-tune an $L$-layer LLM for downstream tasks.

## 3 DEVELOPMENTAL FEDERATED TUNING (DEVFT)

### 3.1 OVERVIEW

Figure 3 illustrates the overall framework of DEVFT, which proceeds through three key steps.

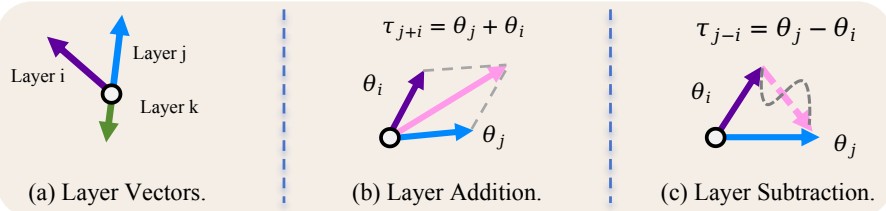

Figure 4: An illustration of layer vectors and layer arithmetic operations.

• **Step ①: Stage Submodel Construction.** Prior to the commencement of each stage, the server constructs a stage-specific submodel. Specifically, the server utilizes the deconfliction-guided layer grouping (DGLG) mechanism (Section 3.2) to cluster layers exhibiting minimal parameter conflicts. Subsequently, the differential-based layer fusion (DBLF) strategy (Section 3.3) is applied to integrate intra-group information, generating a representative layer for each group. Finally, these representative layers are concatenated sequentially to assemble the stage-specific submodel.

• **Step ②: Collaborative Optimization.** Once the submodel is constructed, the federated fine-tuning process commences, wherein devices collaboratively train the submodel on their local data.

• **Step ③: Knowledge Transfer.** Upon completion of the current stage, the acquired knowledge is synchronized to update the global model and is seamlessly transferred to initialize the submodel for the subsequent stage (Section 3.4). This progressive model training process continues until the completion of the $S$-th stage.

## 3.2 DECONFLICTION-GUIDED LAYER GROUPING

As shown in Figure 4(a), parameters of each layer can be represented as corresponding layer vectors, with varying degrees of parameter conflict across layers. When constructing representative layers for each group, strong parameter conflicts between layers can lead to severe information loss, as parameters with opposite signs may neutralize each other's unique contributions during the layer fusion process. This cancellation results in a low-fidelity representative layer that fails to capture the distinct functions of the original layers. Therefore, ensuring high intra-group similarity is critical to minimize this loss and preserve representational fidelity. To achieve this, we propose a deconfliction-guided layer grouping (DGLG) mechanism that clusters layers with minimal parameter conflicts into the same group to preserve their respective knowledge. Specifically, the server initially calculates inter-layer parameter similarity using Equation 1:

$$\text{sim}(\theta_i, \theta_j) = \frac{\langle \theta_i, \theta_j \rangle}{\|\theta_i\| \|\theta_j\|}, \tag{1}$$

where $\theta_i$ and $\theta_j$ denote the parameters of layers $i$ and $j$, respectively, including their associated LoRA parameters. This computation yields a layer similarity matrix $\mathbf{W}$, where each entry $w_{ij}$ measures the parameter similarity between layers $i$ and $j$. Higher similarity values indicate lower parameter conflicts, suggesting these layers should be grouped together. Conversely, lower similarity values signify more severe parameter conflicts, necessitating the assignment of these layers to different groups. Based on the similarity matrix $\mathbf{W}$, we construct a complete undirected graph $G = (\mathcal{V}, \mathcal{E})$, where $\mathcal{V} = \{v_1, v_2, ..., v_L\}$ represents the set of layers and $\mathcal{E} = \{\text{sim}(v_i, v_j)|v_i, v_j \in \mathcal{V}, w_{ij} = w_{ji}\}$ denotes the set of edges weighted by layer similarities. The objective is to partition graph $G$ into $L_s$ non-overlapping groups $\{g_n\}_{n=1}^{L_s}$ for stage $s$, which can be formally expressed as:

$$\min_{\{g_1, g_2, ..., g_{L_s}\}} \sum_{n=1}^{L_s} \sum_{m \neq n} \text{cut}(g_n, g_m), \text{where } \text{cut}(g_n, g_m) = \sum_{p \in g_n} \sum_{q \in g_m} w_{pq},$$

$$\text{s.t. } \forall m, n \in \{1, 2, ..., L_s\}, m \neq n \Rightarrow g_m \cap g_n = \emptyset \text{ and } \bigcup_{n=1}^{L_s} g_n = \mathcal{V}. \tag{2}$$

To solve the optimization problem in Equation 2, we first construct the degree matrix $\mathbf{D} = \text{diag}(d_1, ..., d_L)$, where $d_i = \sum_{j=1}^{L} w_{ij}$ denotes the sum of weights connected to vertex $v_i$. We

then compute the Laplacian matrix as $\mathbf{L} = \mathbf{D} - \mathbf{W}$ and perform eigenvalue decomposition on $\mathbf{L}$ to obtain the eigenvectors corresponding to the $L_s$ smallest eigenvalues. These eigenvectors are stacked column by column to form the embedding matrix $\mathbf{E} \in \mathbb{R}^{L \times L_s}$. Finally, k-means clustering is applied to $\mathbf{E}$ to partition the vertex set into $L_s$ disjoint groups. This process can be formulated as:

$$\{\mathrm{g}_1, \ldots, \mathrm{g}_{L_s}\} = \text{k-means}\left(\mathbf{E}, L_s\right), \quad \mathbf{E} = [\mathbf{v}_1, \ldots, \mathbf{v}_{L_s}],$$

$$\text{where} \quad \mathbf{L} = \mathbf{D} - \mathbf{W}, \quad \mathbf{D} = \text{diag}\left(\sum_{j=1}^{L} w_{1j}, \ldots, \sum_{j=1}^{L} w_{Lj}\right), \tag{3}$$

$$\mathbf{L}\mathbf{v}_t = \lambda_t \mathbf{v}_t, \quad \forall t \in \{1, \ldots, L_s\}, \quad \text{s.t.} \quad \lambda_1 \leq \lambda_2 \leq \cdots \leq \lambda_{L_s},$$

where $\lambda_t$ and $\mathbf{v}_t$ represent the $t$-th eigenvalue and corresponding eigenvector of $\mathbf{L}$. Through this deconfliction-guided layer grouping mechanism, we can partition the $L$ layers of the global model into $L_s$ groups $\{\mathrm{g}_n\}_{n=1}^{L_s}$, where layers within each group exhibit minimal parameter conflicts.

## 3.3 DIFFERENTIAL-BASED LAYER FUSION

After obtaining the partitioned groups, we construct a representative layer for each group. To effectively synthesize these representative layers, we introduce the differential-based layer fusion (DBLF) strategy, which integrates layer information within each group through well-defined arithmetic operations. As illustrated in Figure 4(b), the layer addition operation merges knowledge from two layers, producing a composite layer that encapsulates the semantic information of both source components. Figure 4(c) shows the layer subtraction operation, which distills the unique information present in one layer relative to another. For any given layers $i$ and $j$, these operations are defined as follows:

$$\begin{aligned} \tau_{j+i} &= \theta_j + \theta_i, \\ \tau_{j-i} &= \theta_j - \theta_i, \end{aligned} \tag{4}$$

where $\tau_{j+i}$ and $\tau_{j-i}$ denote the resulting parameter vectors after addition and subtraction operations, respectively. These operations enable fine-grained knowledge editing in the parameter space. A naive approach for intra-group information integration involves performing the addition operation on all layers. However, this introduces significant information redundancy, as layers within the same group $\mathrm{g}_n$ typically share similar functional characteristics. This redundancy limits the submodel's capability to capture diverse and meaningful representations.

To address this challenge, instead of indiscriminately merging all information, DBLF selectively integrates the unique semantic information of each layer. Specifically, it designates the first layer of each group as the *anchor layer* and computes the information differentials of other layers relative to this *anchor layer* through the layer subtraction operation. During layer fusion, only the information differentials are encapsulated into the *anchor layer*, thereby effectively preserving each layer's essential information while eliminating redundancy. This fusion process can be formulated as:

$$\vartheta^{\mathrm{g}_n} = \theta_{anchor} + \beta \sum_{j \in \mathrm{g}_n} (\theta_j - \theta_{anchor}), \tag{5}$$

where $\beta$ is a weighting factor, $\theta_{anchor}$ represents the parameters of the anchor layer, and $\vartheta^{\mathrm{g}_n}$ stands for the representative layer of group $\mathrm{g}_n$, encapsulating the distinctive features of all layers within the group. These derived representative layers are then concatenated sequentially to construct the stage-specific submodel for federated fine-tuning.

## 3.4 KNOWLEDGE TRANSFER

Cross-stage knowledge transfer is critical for cultivating high-performance LLMs, analogous to human cognition where new knowledge builds upon established foundations. At each stage, the acquired knowledge provides optimized initialization for the next-stage submodel, thereby accelerating convergence and improving overall performance by avoiding poor local minima. Through strategic layer clustering and representative layer construction, the encoded knowledge in $\{\vartheta^{\mathrm{g}_n}\}_{n=1}^{L_s}$ can be directly utilized to update all layers within their respective groups $\{\mathrm{g}_n\}_{n=1}^{L_s}$ (step 3 in Figure 3). The rationale is that functionally similar layers inherently exhibit similar parameter distributions and learning patterns. Notably, we only update the LoRA parameters of each layer. This transfer process yields an updated global model that serves as the foundation for constructing the next-stage submodel, ensuring seamless knowledge inheritance across stages. A practical implementation example is detailed in Appendix A.

Table 1: Performance evaluation of **DEVFT** against baseline methods on instruction tuning tasks. **Bold** and underlined values denote the best and second-best results, respectively.

| Method | Close-Ended Benchmark ↑ | | | | | Open-Ended Benchmark ↑ | | | |
|---|---|---|---|---|---|---|---|---|---|
| | TruthfulQA | MMLU | IFEval | BBH | **Average** | Vicuna | MT-1 | MT-2 | **Average** |
| **LLaMA2-7B (INT4)** (Touvron et al., 2023) | | | | | | | | | |
| FedIT | 47.57 | 42.45 | 31.76 | 39.28 | 40.27 | 8.18 | 4.77 | 1.98 | 4.98 |
| DoFIT | 48.32 | 43.04 | 32.62 | 39.59 | 40.89 | 8.19 | 4.92 | 2.13 | 5.08 |
| C2A | 46.71 | 41.83 | 29.45 | 36.07 | 38.52 | 7.66 | 3.97 | 1.88 | 4.50 |
| ProgFed | 48.60 | 43.14 | 32.54 | 39.73 | 41.00 | 8.20 | 4.88 | 2.19 | 5.09 |
| FLoRA | 47.76 | 42.64 | 32.08 | 39.25 | 40.43 | 8.21 | 4.85 | 2.02 | 5.03 |
| FedSA-LoRA | 48.24 | 42.91 | 32.71 | 39.36 | 40.81 | 8.26 | 5.09 | 2.31 | 5.22 |
| DEVFT | **50.28** | **44.15** | **33.97** | **40.93** | **42.33** | **8.41** | **5.76** | **2.92** | **5.70** |
| **LLaMA3.1-8B (INT4)** (Grattafiori et al., 2024) | | | | | | | | | |
| FedIT | 48.07 | 63.31 | 47.32 | 62.69 | 55.35 | 8.89 | 6.54 | 5.03 | 6.82 |
| DoFIT | 49.12 | 65.17 | 51.66 | 65.21 | 57.79 | 9.01 | 6.72 | 5.22 | 6.98 |
| C2A | 48.99 | 63.76 | 46.10 | 61.85 | 55.18 | 8.74 | 6.67 | 4.98 | 6.80 |
| ProgFed | 53.12 | 66.77 | 54.55 | 66.03 | 60.12 | 9.07 | 6.85 | 5.08 | 7.00 |
| FLoRA | 50.23 | 64.95 | 50.47 | 64.93 | 57.65 | 8.96 | 6.75 | 5.11 | 6.94 |
| FedSA-LoRA | 53.29 | 66.87 | 56.17 | 67.56 | 60.97 | 9.03 | 6.92 | 5.41 | 7.12 |
| DEVFT | **55.23** | **68.42** | **62.29** | **71.04** | **64.25** | **9.18** | **7.63** | **6.57** | **7.79** |
| **LLaMA2-13B (INT4)** (Touvron et al., 2023) | | | | | | | | | |
| FedIT | 52.40 | 55.45 | 40.33 | 46.14 | 48.58 | 8.37 | 5.17 | 3.01 | 5.52 |
| DoFIT | 54.77 | 56.09 | 41.68 | 46.41 | 49.74 | 8.37 | 5.19 | 3.34 | 5.63 |
| C2A | 53.91 | 54.33 | 38.96 | 45.06 | 48.07 | 8.05 | 5.08 | 3.26 | 5.46 |
| ProgFed | 55.01 | 57.38 | 42.13 | 46.36 | 50.22 | 8.38 | 5.28 | 3.07 | 5.58 |
| FLoRA | 54.26 | 56.23 | 41.49 | 46.32 | 49.58 | 8.40 | 5.22 | 3.15 | 5.59 |
| FedSA-LoRA | 55.73 | 57.51 | 43.21 | 46.91 | 50.84 | 8.49 | 5.39 | 3.45 | 5.78 |
| DEVFT | **57.19** | **58.74** | **46.45** | **48.70** | **52.77** | **8.67** | **6.18** | **4.52** | **6.46** |

## 4 EXPERIMENTS

### 4.1 EXPERIMENTAL SETUP

**Models and Datasets.** Following OpenFedLLM (Ye et al., 2024), we evaluate DEVFT on three LLaMA-based models: LLaMA2-7B (Touvron et al., 2023), LLaMA3.1-8B (Grattafiori et al., 2024), and LLaMA2-13B. All models are fine-tuned on the Alpaca-GPT4 dataset (Peng et al., 2023), and are evaluated on both close-ended and open-ended benchmarks. Specifically, the close-ended benchmarks include TruthfulQA (Lin et al., 2022), MMLU (Hendrycks et al., 2020), IFEval (Zhou et al., 2023), and BBH (Suzgun et al., 2023), which assess the models' capabilities in honesty and truthfulness, knowledge coverage, instruction following, and reasoning, respectively. The open-ended benchmarks, including Vicuna-Bench (Chiang et al., 2023) and MT-Bench (Zheng et al., 2023), evaluate the models' performance in multi-turn dialogue scenarios.

**Implementation Details.** The fine-tuning process is divided into four stages ($S = 4$) for all models, with each stage's submodel receiving an equal number of federated fine-tuning rounds. The capacity of the submodels doubles at each stage. Specifically, for LLaMA2-7B and LLaMA3.1-8B, the submodel capacities across the four stages are {4, 8, 16, 32}, whereas for LLaMA2-13B, they are {5, 10, 20, 40}. We set the hyperparameter $\beta$ to 0.1 for LLaMA2-7B and LLaMA3.1-8B, and 0.15 for LLaMA2-13B. Additional implementation details are provided in Appendix B.

### 4.2 BASELINES

**Resource-Unaware Methods**. FedIT (Zhang et al., 2024a) integrates LoRA with FedAvg to enable federated instruction tuning. DoFIT (Xu et al., 2024a) employs specialized LoRA initialization and aggregation strategies to mitigate catastrophic forgetting. C2A (Kim et al., 2023) addresses data heterogeneity through a hypernetwork that dynamically generates device-specific adapters.

**Resource-Aware Methods**. ProgFed (Wang et al., 2022) partitions the global model into blocks and gradually incorporates them for training. FLoRA (Wang et al., 2024) allocates different LoRA

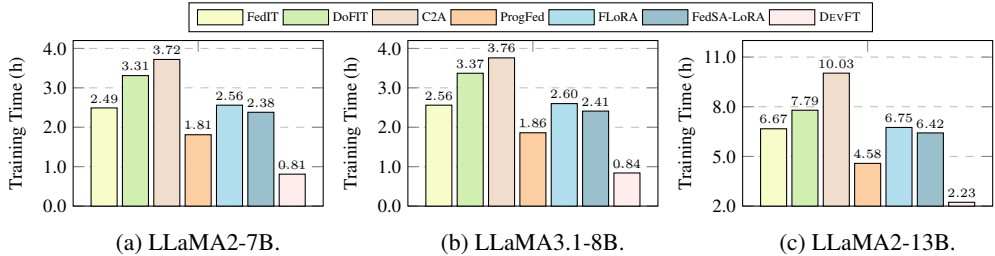

Figure 5: Comparative analysis of cumulative local training time across different methods.

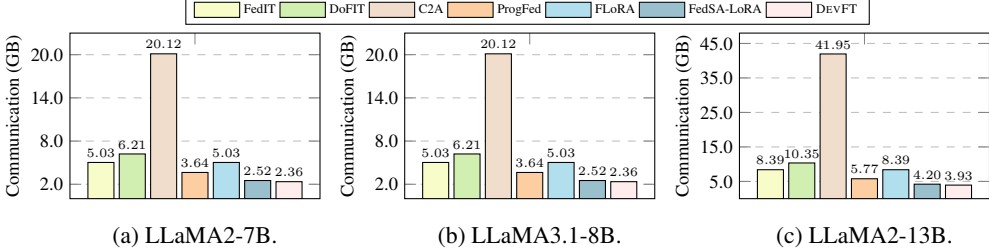

Figure 6: Comparative analysis of total communication overhead across different methods.

ranks to devices based on their resources. FedSA-LoRA (Guo et al., 2025) only shares matrices **A** with the server to reduce resource costs.

## 4.3 PERFORMANCE EVALUATION

Table 1 presents the comprehensive performance comparison. DEVFT consistently outperforms baseline methods across all settings, demonstrating its effectiveness and robustness.

1) **Comparison with Resource-Unaware Methods.** Resource-unaware methods uniformly demonstrate inferior performance. On close-ended benchmarks, FedIT suffers average performance drops of 2.06%, 8.9%, and 4.19% on LLaMA2-7B, LLaMA3.1-8B, and LLaMA2-13B, respectively. For open-ended benchmarks, the gaps are 0.72, 0.97, and 0.94. This degradation primarily arises from noise introduced by FedIT's independent aggregation of matrices **A** and **B**. DoFIT achieves moderate gains through specialized initialization and aggregation strategies but still lags behind DEVFT, with a gap of up to 10.63% on LLaMA3.1-8B. Similarly, C2A performs notably worse than DEVFT, with average performance drops of up to 9.07% and 1.2 in close-ended and open-ended benchmarks, respectively, underscoring the inherent limitations of adapter-based methods.

2) **Comparison with Resource-Aware Methods.** Resource-aware methods generally outperform resource-unaware counterparts, but still fall short of DEVFT. ProgFed shows average performance degradation of 1.33% and 0.61 on LLaMA2-7B, 4.13% and 0.79 on LLaMA3.1-8B, and 2.55% and 0.88 on LLaMA2-13B for close-ended and open-ended benchmarks, respectively. FedSA-LoRA exhibits similar performance degradation patterns to ProgFed, while FLoRA demonstrates more significant performance deterioration. In particular, for close-ended benchmarks, FedSA-LoRA shows average performance decrements ranging from 1.52% to 3.28% across these models, whereas FLoRA exhibits more substantial degradation, with decrements spanning from 1.9% to 6.6%. The superiority of DEVFT stems from its developmental paradigm, which progressively builds a powerful LLM from a compact foundation, effectively preventing convergence to local minima.

## 4.4 EFFICIENCY EVALUATION

In this section, we evaluate the efficiency of DEVFT from both computational and communication perspectives. Furthermore, we present a detailed analysis of training overhead across different developmental stages to understand how DEVFT effectively optimizes resource utilization.

**Computation Efficiency.** Instead of using floating-point operations per second (FLOPs) to evaluate computation efficiency, we employ wall-clock training time to provide a more intuitive reflection of real-world deployment efficiency for each method. Specifically, we measure the cumulative local

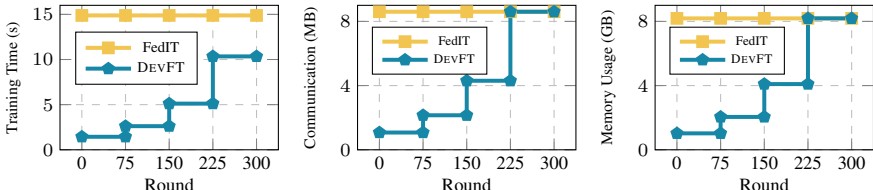

Figure 7: Resource consumption analysis of a device per round: *training time, communication overhead, and memory usage* for LLaMA2-7B.

training time required for each method to reach convergence, with results shown in Figure 5. Our experimental results demonstrate that DEVFT significantly accelerates model convergence across all model architectures. Notably, for LLaMA2-7B, DEVFT achieves up to $4.59\times$ speedup in convergence time. This improvement can be attributed to the developmental training paradigm of DEVFT: early fine-tuning on compact submodels significantly reduces computational overhead, while cross-stage knowledge transfer further expedites convergence when scaling to larger submodels.

**Communication Efficiency.** Figure 6 illustrates the total communication overhead required for each method to reach convergence. DEVFT consistently achieves convergence with minimal communication costs across all experimental settings, reducing communication overhead by up to $10.67\times$ on LLaMA2-13B. This communication efficiency stems from the fact that DEVFT only transmits a small number of LoRA parameters to the server during the initial $S-1$ stages.

**Detailed Overhead Analysis.** To gain a deeper understanding of DEVFT's efficiency, Figure 7 reports the per-round resource consumption on each device for FedIT and DEVFT, including training time, communication overhead, and memory usage. FedIT consistently incurs high resource costs throughout fine-tuning. In contrast, DEVFT exhibits a more efficient pattern, with resource requirements gradually increasing as the submodel capacity expands, thereby substantially reducing overall training overhead. In the early stages, particularly the first stage, DEVFT achieves significant resource savings compared to FedIT, reducing per-round training time by $10.3\times$, communication overhead by $4\times$, and memory usage by $4\times$. Intriguingly, we discover that fine-tuning the reconstructed models of DEVFT at each stage also yields acceleration compared to directly fine-tuning pre-trained models. For example, even in the fourth stage where the submodel grows to match the target model size, DEVFT still delivers a $1.44\times$ speedup per round.

Table 2: Ablation study on different layer grouping strategies.

| Method | Close-Ended Benchmark ↑ | | | | |
|---|---|---|---|---|---|
| | TruthfulQA | MMLU | IFEval | BBH | Average |
| **LLaMA2-7B (INT4)** (Touvron et al., 2023) | | | | | |
| DGLG | **50.28** | **44.15** | **33.97** | **40.93** | **42.33** |
| RANDOM | 47.89 | 42.09 | 29.18 | 38.45 | 39.90 (↓ 2.43) |
| EVEN | 45.41 | 39.83 | 25.04 | 36.73 | 36.25 (↓ 6.08) |
| **LLaMA3.1-8B (INT4)** (Touvron et al., 2023) | | | | | |
| DGLG | **55.23** | **68.42** | **62.29** | **71.04** | **64.25** |
| RANDOM | 51.02 | 66.74 | 54.89 | 70.11 | 60.69 (↓ 3.56) |
| EVEN | 48.51 | 62.50 | 50.01 | 70.03 | 57.76 (↓ 6.49) |

Table 3: Ablation study on different representative layer construction methods.

| Method | Close-Ended Benchmark ↑ | | | | |
|---|---|---|---|---|---|
| | TruthfulQA | MMLU | IFEval | BBH | Average |
| **LLaMA2-7B (INT4)** (Touvron et al., 2023) | | | | | |
| DBLF | **50.28** | **44.15** | **33.97** | **40.93** | **42.33** |
| R-ONE | 46.75 | 40.13 | 26.38 | 37.62 | 37.72 (↓ 4.61) |
| SUM | 48.15 | 42.91 | 30.69 | 39.84 | 40.90 (↓ 1.43) |
| **LLaMA3.1-8B (INT4)** (Touvron et al., 2023) | | | | | |
| DBLF | **55.23** | **68.42** | **62.29** | **71.04** | **64.25** |
| R-ONE | 47.51 | 57.33 | 50.21 | 58.09 | 53.29 (↓ 10.96) |
| SUM | 52.74 | 65.18 | 58.47 | 68.39 | 61.20 (↓ 3.05) |

## 4.5 ABLATION STUDY

**Effect of the Deconfliction-Guided Layer Grouping Mechanism.** To understand the significance of the deconfliction-guided layer grouping (DGLG), we compare it with two baselines: random grouping (RANDOM) and even grouping (EVEN). Table 2 shows that DGLG consistently outperforms both baselines across all settings. For LLaMA2-7B, RANDOM and EVEN incur average performance drops of 2.43% and 6.08%, respectively, relative to DGLG. Similar trends are observed on LLaMA3.1-8B, where the degradations reach 3.56% and 6.49%, respectively. These results demonstrate that DGLG effectively enhances the layer fusion process by clustering layers with minimal parameter conflicts into the same group, thereby preserving more informative representations.

Table 4: Evaluation of DEVFT's compatibility with existing methods.

| Method | Close-Ended Benchmark ↑ | | | | | Resource ↓ | |
|---|---|---|---|---|---|---|---|
| | TruthfulQA | MMLU | IFEval | BBH | Average | Time (h) | Comm. (GB) |
| **LLaMA2-7B (INT4)** (Touvron et al., 2023) | | | | | | | |
| FedIT | 47.57 | 42.45 | 31.76 | 39.28 | 40.27 | 2.49 | 5.03 |
| FedIT+DEVFT | **49.86** | **43.87** | **33.65** | **40.79** | **42.04** (↑ 1.77) | **0.83** (×3.00) | **2.36** (×2.13) |
| FedSA-LoRA | 48.24 | 42.91 | 32.71 | 39.36 | 40.81 | 2.38 | 2.52 |
| FedSA-LoRA+DEVFT | **50.42** | **44.57** | **40.92** | **41.36** | **44.32** (↑ 3.51) | **0.72** (×3.31) | **1.18** (×2.14) |
| **LLaMA2-13B (INT4)** (Touvron et al., 2023) | | | | | | | |
| FedIT | 52.40 | 55.45 | 40.33 | 46.14 | 48.58 | 6.67 | 8.39 |
| FedIT+DEVFT | **56.84** | **58.26** | **45.49** | **48.52** | **52.28** (↑ 3.70) | **2.30** (×2.90) | **3.93** (×2.13) |
| FedSA-LoRA | 55.73 | 57.51 | 43.21 | 46.91 | 50.84 | 6.42 | 4.20 |
| FedSA-LoRA+DEVFT | **57.61** | **59.25** | **47.63** | **49.13** | **53.41** (↑ 2.57) | **2.19** (×2.93) | **1.97** (×2.13) |

Table 5: Performance analysis of different initial submodel capacities.

| Initial | Close-Ended Benchmark ↑ | | | | |
|---|---|---|---|---|---|
| Capacity | TruthfulQA | MMLU | IFEval | BBH | Average |
| **LLaMA3.1-8B (INT4)** (Touvron et al., 2023) | | | | | |
| 1 | 52.45 | 66.85 | 56.83 | 70.12 | 61.56 (↓ 2.69) |
| 2 | 53.87 | 67.31 | 59.45 | 70.50 | 62.78 (↓ 1.47) |
| 4 | **55.23** | **68.42** | **62.29** | **71.04** | **64.25** |
| 8 | 53.21 | 67.12 | 58.35 | 70.65 | 62.33 (↓ 1.92) |
| 16 | 51.08 | 65.89 | 54.12 | 70.01 | 60.28 (↓ 3.97) |
| 32 | 48.79 | 64.49 | 49.75 | 69.33 | 58.09 (↓ 6.16) |

Table 6: Performance analysis under varying submodel growth rates.

| Growth | Close-Ended Benchmark ↑ | | | | |
|---|---|---|---|---|---|
| Rate | TruthfulQA | MMLU | IFEval | BBH | Average |
| **LLaMA2-7B (INT4)** (Touvron et al., 2023) | | | | | |
| 2 | **50.28** | **44.15** | **33.97** | **40.93** | **42.33** |
| 4 | 47.96 | 42.56 | 29.87 | 38.79 | 39.80 (↓ 2.53) |
| 8 | 45.68 | 40.07 | 25.63 | 36.92 | 37.08 (↓ 5.25) |
| **LLaMA2-13B (INT4)** (Touvron et al., 2023) | | | | | |
| 2 | **57.19** | **58.74** | **46.45** | **48.70** | **52.77** |
| 4 | 52.23 | 56.78 | 34.56 | 42.29 | 46.47 (↓ 6.3) |
| 8 | 48.12 | 52.33 | 26.78 | 37.45 | 41.17 (↓ 11.6) |

**Effect of the Differential-Based Layer Fusion Strategy.** To evaluate the effectiveness of the differential-based layer fusion (DBLF), we compare it against two baselines: R-ONE, which randomly selects one layer from each group as the representative layer, and SUM, which directly performs the addition operation on all layers within each group to generate the representative layer. As shown in Table 3, DBLF consistently outperforms both baselines. On LLaMA2-7B, R-ONE and SUM incur average performance drops of 4.61% and 1.43%, respectively, relative to DBLF. The performance gap widens on LLaMA3.1-8B, with degradations reaching 10.96% for R-ONE and 3.05% for SUM. These results confirm that DBLF can effectively capture and integrate the unique semantic information from layers within each group, leading to superior representative layer construction.

## 4.6 ANALYSIS

**Compatibility with Existing Methods.** We further validate the compatibility of DEVFT with existing approaches by integrating it with FedIT and FedSA-LoRA. Table 4 shows that incorporating DEVFT consistently improves model performance and system efficiency. For example, combining DEVFT with FedIT on LLaMA2-13B yields a 3.7% average performance gain, 2.9× faster convergence, and a 2.13× reduction in communication overhead. Similar improvements are observed when integrating DEVFT with FedSA-LoRA. These results indicate that DEVFT functions as a general framework that seamlessly enhances existing methods while preserving their inherent strengths.

**Impact of Initial Submodel Capacity.** We also conduct experiments to investigate how the initial capacity of submodels influences the overall model performance. Specifically, we experiment with LLaMA3.1-8B and set different initial capacities {1,2,4,8,16,32}, while maintaining the same total training budget. The submodel capacity also doubles progressively until reaching the full model capacity. Table 5 shows that the model achieves optimal performance when the initial capacity is set to 4, while either smaller or larger initial capacities result in performance degradation. This phenomenon is analogous to human learning, where starting from either too early (infancy) or too late (adulthood) may lead to suboptimal outcomes due to premature or delayed cognitive development.

**Impact of Submodel Growth Rate.** We explore how different submodel growth rates affect overall performance. Specifically, we experiment with diverse capacity scaling multipliers {2,4,8}. For instance, a multiplier of 4 indicates that the submodel capacity quadruples at each stage until reaching the full capacity. This generates capacity sequences of {4→16→32} for LLaMA2-7B

Table 7: Scalability and robustness analysis of DevFT on text classification tasks.

| Method | Evaluation Benchmark ↑ | | | | |
|---|---|---|---|---|---|
| | YELP-P | AGNEWS | YAHOO | 20NEWS | **Average** |
| **BERT (Devlin et al., 2019)** | | | | | |
| FedIT | 83.12 | 87.05 | 68.34 | 76.89 | 78.85 |
| DEVFT | 84.53 | 90.91 | 70.67 | 80.06 | **81.54** (↑2.69) |
| **RoBERTa (Liu et al., 2019)** | | | | | |
| FedIT | 82.93 | 87.86 | 68.21 | 77.32 | 79.08 |
| DEVFT | 84.02 | 90.27 | 71.35 | 79.83 | **81.37** (↑2.29) |

and $\{5\rightarrow20\rightarrow40\}$ for LLaMA2-13B. Table 6 demonstrates that higher growth rates significantly compromise model performance. For LLaMA2-7B, scaling multipliers of 4 and 8 lead to average performance drops of 2.53% and 5.25%, respectively. The degradation is even more pronounced on LLaMA2-13B, with decreases of 6.3% and 11.6%. This deterioration can be attributed to abrupt capacity transitions, which may disrupt the construction of the knowledge structure. This phenomenon mirrors natural learning processes, where steady, incremental development typically yields better long-term outcomes compared to the aggressive pursuit of short-term performance gains.

**Scalability and Robustness Analysis.** To assess DEVFT's scalability across varying device populations and its robustness to data heterogeneity, we conduct additional experiments on text classification tasks partitioned via a Dirichlet distribution ($\alpha = 1$). Our experimental setup spans a wide range of scales, including 100 devices for 20NEWS (Lang, 1995), 1,000 for AGNEWS (Zhang et al., 2015) and YELP-P (Zhang et al., 2015), and up to 10,000 devices for YAHOO (Zhang et al., 2015), utilizing BERT (Devlin et al., 2019) and RoBERTa (Liu et al., 2019) as global models. The results in Table 7 indicate that DEVFT consistently outperforms the baseline in these heterogeneous settings, achieving an average performance gain of up to 2.69% on BERT. This success underscores the scalability and robustness of our method, which stems from our developmental training paradigm: by initiating training with smaller, less complex submodels, DEVFT effectively mitigates the risk of client drift and overfitting to local data distributions during the critical early phases.

# 5 FUTURE WORK

The developmental analogy underlying DEVFT can be interpreted through two distinct lenses: a Capacity Curriculum (mirroring the physical development of the brain) and a Data Curriculum (mirroring the progression from simple to complex knowledge) (Soviany et al., 2022). In this work, DEVFT strategically prioritizes the Capacity Curriculum to address the immediate and prohibitive hardware bottlenecks prevalent in resource-constrained federated edge environments. However, we recognize that integrating a Data Curriculum represents a logical and promising evolution of the framework. Future research could explore synergizing model-level growth with data-level curriculum learning—for instance, feeding simpler instructions to early-stage submodels while reserving complex reasoning tasks for the fully matured model—to fully realize the cognitive metaphor.

# 6 CONCLUSION

In this paper, we introduce DEVFT, an innovative federated fine-tuning framework that substantially reduces the resource consumption of LLM adaptation through cognitive developmental training. DEVFT decomposes the fine-tuning process into several developmental stages, where each stage adapts a submodel with increasing parameter capacity. To efficiently architect these stage-specific submodels, DEVFT integrates two key techniques: a deconfliction-guided layer grouping mechanism and a differential-based layer fusion strategy. Extensive evaluations across multiple benchmarks demonstrate that DEVFT achieves superior performance with significantly enhanced efficiency. Moreover, it maintains high compatibility with existing federated fine-tuning methods, offering a robust and versatile enhancement to the current ecosystem.

ACKNOWLEDGEMENTS

This work is supported in part by the Science and Technology Development Fund of Macau (0107/2024/RIA2), Joint Science and Technology Research Project with Hong Kong and Macau in Key Areas of Nansha District's Science and Technology Plan (EF2024-00180-IOTSC), and the Multi-Year Research Grant of University of Macau (MYRG-GRG2023-00211-IOTSC-UMDF and MYRG-GRG2024-00180-IOTSC).

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

## A  A CONCRETE EXAMPLE OF DEVFT IN PRACTICE

Assume the full model consists of six layers: $[\theta_1, \theta_2, \theta_3, \theta_4, \theta_5, \theta_6]$, and DEVFT proceeds in three stages with sub-model capacities: $\{2 \rightarrow 4 \rightarrow 6\}$.

**Stage 1 (Capacity = 2):**

- DGLG partitions the layers into two groups: $[\{\theta_1, \theta_2, \theta_3\}, \{\theta_4, \theta_5, \theta_6\}]$.
- DBLF produces two representative layers: $\{\theta_1, \theta_2, \theta_3\} \rightarrow \vartheta^{g_1}; \{\theta_4, \theta_5, \theta_6\} \rightarrow \vartheta^{g_2}$.
- These two representative layers form the sub-model $[\vartheta^{g_1}, \vartheta^{g_2}]$ for the current stage.
- After completing the current stage, we perform knowledge transfer as follows: $\vartheta^{g_1} \rightarrow \theta_{1,2,3}; \vartheta^{g_2} \rightarrow \theta_{4,5,6}$.
- All layers are updated, and the resulting global model then serves as the foundation for constructing the submodel in the next stage.

**Stage 2 (Capacity = 4):**

- DGLG partitions the layers into four groups: $[\{\theta_1\}, \{\theta_2, \theta_3\}, \{\theta_4, \theta_5\}, \{\theta_6\}]$.
- DBLF produces four representative layers: $\{\theta_1\} \rightarrow \vartheta^{g_1}; \{\theta_2, \theta_3\} \rightarrow \vartheta^{g_2}; \{\theta_4, \theta_5\} \rightarrow \vartheta^{g_3}; \{\theta_6\} \rightarrow \vartheta^{g_4}$.
- These four representative layers form the sub-model $[\vartheta^{g_1}, \vartheta^{g_2}, \vartheta^{g_3}, \vartheta^{g_4}]$ for the current stage.
- After completing the current stage, we perform knowledge transfer as follows: $\vartheta^{g_1} \rightarrow \theta_1; \vartheta^{g_2} \rightarrow \theta_{2,3}; \vartheta^{g_3} \rightarrow \theta_{4,5}; \vartheta^{g_4} \rightarrow \theta_6$.
- All layers are updated, and the resulting global model then serves as the foundation for constructing the submodel in the next stage.

**Stage 3 (Capacity = 6):**

- The submodel now encompasses all six layers, reaching the model's full capacity and enabling end-to-end fine-tuning of the global model to complete the training process.

This example serves as a simplified illustration to intuitively demonstrate the multi-stage grouping concept. Empirically, we observe that neighboring layers often exhibit high similarity in both function and parameters due to the hierarchical nature of information processing in Transformers. Consequently, these layers are frequently clustered together by DGLG. Thus, while DGLG operates independently of layer positions, this contiguous grouping serves as a representative and intuitive visualization of the actual process.

## B  ADDITIONAL IMPLEMENTATION DETAILS

Our DEVFT is implemented using PyTorch with the support of HuggingFace Transformers library (Wolf et al., 2019) for model and dataset management. Following the experimental setup of OpenFedLLM (Ye et al., 2024), we randomly distribute the Alpaca-GPT4 dataset across 20 devices, with 10% of devices randomly sampled for participation in each training round. Each selected device performs 10 local training iterations with a batch size of 16. The local fine-tuning process utilizes the AdamW optimizer coupled with a cosine learning rate scheduler. We adopt a staged learning rate strategy, starting at 1e-6 and incrementing by a factor of 10 at each subsequent stage until reaching 1e-4. Additionally, we exclusively apply LoRA to $\mathbf{W}_q$ and $\mathbf{W}_v$ matrices in the attention layers (Xu et al., 2026a;b) and configure the LoRA module with a rank of 32. The maximum sequence length is set to 512 tokens (Ye et al., 2024). The total number of federated fine-tuning rounds is set to 300 for LLaMA2-7B and LLaMA3.1-8B, and increases to 400 for LLaMA2-13B. Moreover, to improve computational efficiency, we apply INT4 quantization (Ye et al., 2024) to all models and conduct experiments on a single NVIDIA H800 GPU. To ensure the reliability of our results, all experiments are repeated multiple times, with the averaged values reported as the final results.

## C Theoretical Convergence Analysis

We analyze DEVFT under standard nonconvex FL assumptions (Li et al., 2019; Wang et al., 2022; Wu et al., 2025c). The analysis is aligned with the actual pipeline: at each stage, the current full model is regrouped, DBLF constructs representative-layer initialization, the stage submodel is trained by FedAvg, and the trained submodel is mapped back to the full model by group-wise replication.

### C.1 Preliminaries and Stage Transition

**Full objective.** Let $\Theta \in \mathbb{R}^d$ denote full-model trainable parameters ($L$ layers). For client $i \in [N]$:

$$f_i(\Theta) = \mathbb{E}_{\xi \sim \mathcal{D}_i}[\ell(\Theta; \xi)], \qquad f(\Theta) = \frac{1}{N} \sum_{i=1}^{N} f_i(\Theta), \qquad f_{\inf} := \inf_{\Theta} f(\Theta) > -\infty. \tag{6}$$

Use block norm $\|\Theta\|^2 = \sum_{j=1}^{L} \|\theta_j\|^2$.

**Stage-wise dynamic grouping.** At stage $s \in [S]$, grouping is recomputed from the previous fused full model $\Theta_{\text{fuse}}^{(s-1)}$ (for $s = 1$, $\Theta_{\text{fuse}}^{(0)}$ is the given initialization):

$$\Pi_s = \{g_n^{(s)}\}_{n=1}^{L_s} = \text{DGLG}\big(\Theta_{\text{fuse}}^{(s-1)}\big), \qquad L_1 < \cdots < L_S = L. \tag{7}$$

Let $m_s := \max_n |g_n^{(s)}|$ and $d_s = L_s d_{\text{layer}}$.

**Stage-start DBLF construction (full → submodel).** For each group $g_n^{(s)}$ with anchor $a(n)$, define representative block

$$u_{0,n}^{(s)} = \theta_{\text{fuse},a(n)}^{(s-1)} + \beta \sum_{j \in g_n^{(s)}} \left( \theta_{\text{fuse},j}^{(s-1)} - \theta_{\text{fuse},a(n)}^{(s-1)} \right), \qquad \beta \in (0, 1]. \tag{8}$$

Stacking $\{u_{0,n}^{(s)}\}_{n=1}^{L_s}$ gives

$$u_0^{(s)} = (u_{0,1}^{(s)}, \ldots, u_{0,L_s}^{(s)}) \in \mathbb{R}^{d_s}. \tag{9}$$

**Stage embedding and objective.** Define replication embedding $\mathcal{E}_s^{\Pi_s} : \mathbb{R}^{d_s} \to \mathbb{R}^d$: for any $j \in g_n^{(s)}$, $[\mathcal{E}_s^{\Pi_s}(u)]_j = u_n$. Equivalently, $\mathcal{E}_s^{\Pi_s}(u) = A_s u$, where

$$\|A_s\|_2^2 = m_s. \tag{10}$$

Define stage objective

$$F_s(u; \Pi_s) := \frac{1}{N} \sum_{i=1}^{N} f_i\big(\mathcal{E}_s^{\Pi_s}(u)\big), \qquad F_s^*(\Pi_s) := \inf_u F_s(u; \Pi_s). \tag{11}$$

If each $f_i$ is $L_f$-smooth in $\Theta$, then $F_s(\cdot; \Pi_s)$ is $(L_f m_s)$-smooth. Let

$$\bar{L} := L_f \max_{s \in [S]} m_s, \tag{12}$$

so every stage objective is $\bar{L}$-smooth.

**Stage-end map-back (submodel → full).** After stage-$s$ training obtains $u_{T_s}^{(s)}$, define

$$\Theta_{\text{fuse}}^{(s)} := \mathcal{E}_s^{\Pi_s}\left(u_{T_s}^{(s)}\right). \tag{13}$$

This is pure replication (broadcast) and introduces no extra fusion perturbation.

**Federated protocol.** Each round samples $M = qN$ clients ($q \in (0, 1)$), each selected client performs $K$ local SGD steps (stepsize $\eta$), then server averages (FedAvg (McMahan et al., 2017)).

**Assumptions.**

**Assumption C.1** (Smoothness). $\|\nabla f_i(\Theta) - \nabla f_i(\Theta')\| \leq L_f \|\Theta - \Theta'\|$, $\forall i, \Theta, \Theta'$.

**Assumption C.2** (Unbiased stochastic gradients, bounded variance). $\mathbb{E}[g_i(\Theta; \xi)] = \nabla f_i(\Theta)$ and $\mathbb{E}\|g_i(\Theta; \xi) - \nabla f_i(\Theta)\|^2 \leq \sigma^2$.

**Assumption C.3** (Bounded heterogeneity). $\frac{1}{N} \sum_{i=1}^{N} \|\nabla f_i(\Theta)\|^2 \leq \|\nabla f(\Theta)\|^2 + G^2$, $\forall \Theta$.

## C.2 PER-STAGE CONVERGENCE

**Theorem C.4** (Per-stage stationarity). *Fix stage $s$ and condition on grouping $\Pi_s$. Under Assumptions C.1–C.3, run FedAvg on $F_s(\cdot; \Pi_s)$ for $T_s$ rounds, $K$ local steps each round, and $M = qN$ participants each round. If $\eta_s \leq \frac{1}{4\bar{L}K}$, then*

$$\frac{1}{T_s} \sum_{t=0}^{T_s-1} \mathbb{E}\Big[\|\nabla F_s(u_t^{(s)}; \Pi_s)\|^2 \mid \Pi_s\Big] \leq \frac{2\Big(F_s(u_0^{(s)}; \Pi_s) - F_s^*(\Pi_s)\Big)}{\eta_s K T_s} + \frac{\bar{L}\,\eta_s\,\sigma^2}{qN} + c_1 \bar{L}^2 \eta_s^2 K^2 G^2, \tag{14}$$

*where $c_1 > 0$ is an absolute constant.*

**Corollary C.5** (Per-stage $O(1/\sqrt{T_s})$). *If $\eta_s = \frac{c_0}{K\sqrt{T_s}}$ with $c_0 \leq \frac{1}{4\bar{L}}$, then*

$$\frac{1}{T_s} \sum_{t=0}^{T_s-1} \mathbb{E}\Big[\|\nabla F_s(u_t^{(s)}; \Pi_s)\|^2 \mid \Pi_s\Big] \leq O\left(\frac{F_s(u_0^{(s)}; \Pi_s) - F_s^*(\Pi_s)}{\sqrt{T_s}}\right) + O\left(\frac{1}{qN K\sqrt{T_s}}\right) + O\left(\frac{1}{T_s}\right). \tag{15}$$

## C.3 DBLF CONSTRUCTION PERTURBATION BOUND

DBLF perturbation occurs at *stage start* (Equation 8).

**Lemma C.6** (Stage-start DBLF perturbation). *Define the pre-construction intra-group diameter*

$$\delta_{s-1}^{\text{pre}} := \max_n \max_{j,k \in \text{g}_n^{(s)}} \left\|\theta_{\text{fuse},j}^{(s-1)} - \theta_{\text{fuse},k}^{(s-1)}\right\|. \tag{16}$$

*Then for any group $\text{g}_n^{(s)}$ and any $j \in \text{g}_n^{(s)}$,*

$$\left\|u_{0,n}^{(s)} - \theta_{\text{fuse},j}^{(s-1)}\right\| \leq \Big(1 + \beta(|\text{g}_n^{(s)}| - 1)\Big)\delta_{s-1}^{\text{pre}} \leq (1 + \beta m_s)\delta_{s-1}^{\text{pre}}. \tag{17}$$

*Let $\widetilde{\Theta}_0^{(s)} := \mathcal{E}_s^{\Pi_s}(u_0^{(s)})$. Then*

$$\left\|\widetilde{\Theta}_0^{(s)} - \Theta_{\text{fuse}}^{(s-1)}\right\| \leq \sqrt{L}\,(1 + \beta m_s)\delta_{s-1}^{\text{pre}}. \tag{18}$$

*Proof.* From Equation 8,

$$u_{0,n}^{(s)} - \theta_{\text{fuse},j}^{(s-1)} = \Big(\theta_{\text{fuse},a(n)}^{(s-1)} - \theta_{\text{fuse},j}^{(s-1)}\Big) + \beta \sum_{k \in \text{g}_n^{(s)}} \Big(\theta_{\text{fuse},k}^{(s-1)} - \theta_{\text{fuse},a(n)}^{(s-1)}\Big). \tag{19}$$

Applying triangle inequality and the definition of $\delta_{s-1}^{\text{pre}}$ gives Equation 17. Summing squared block deviations over $L$ layers yields Equation 18. $\square$

## C.4 FINAL-STAGE FULL-OBJECTIVE GUARANTEE

At stage $S$, $L_S = L$, and thus $F_S(\cdot; \Pi_S) \equiv f(\cdot)$. By construction,

$$\Theta_0^{(S)} = \Theta_{\text{fuse}}^{(S-1)}. \tag{20}$$

Also define the DBLF-replicated start point

$$\widetilde{\Theta}_0^{(S)} := \mathcal{E}_S^{\Pi_S}(u_0^{(S)}), \tag{21}$$

Table 8: Assessing the generalization capability of DEVFT in centralized training scenarios.

| Method | Close-Ended Benchmark ↑ | | | | | Time (h) ↓ |
|--------|-----------|------|--------|-----|---------|------------|
| | TruthfulQA | MMLU | IFEval | BBH | Average | |
| LLaMA2-7B (INT4) (Touvron et al., 2023) | | | | | | |
| End-to-End | 48.36 | 43.27 | 32.64 | 39.54 | 40.95 | 1.77 |
| DEVFT | **52.39** | **47.59** | **35.86** | **42.25** | **44.52** (↑ 3.57) | **0.55** (×3.22) |
| LLaMA2-13B (INT4) (Touvron et al., 2023) | | | | | | |
| End-to-End | 54.68 | 57.32 | 42.51 | 46.98 | 50.37 | 4.28 |
| DEVFT | **59.38** | **61.82** | **48.57** | **49.95** | **54.93** (↑ 4.56) | **1.41** (×3.04) |

for which Lemma C.6 gives

$$\|\widetilde{\Theta}_0^{(S)} - \Theta_0^{(S)}\| \leq \sqrt{L}\,(1 + \beta m_S)\delta_{S-1}^{\mathrm{pre}}. \tag{22}$$

Using $L_f$-smoothness and Young's inequality, for any $\alpha > 0$:

$$f(\Theta + \Delta) \leq f(\Theta) + \frac{1}{2\alpha}\|\nabla f(\Theta)\|^2 + \frac{\alpha + L_f}{2}\|\Delta\|^2. \tag{23}$$

**Theorem C.7** (Final-stage convergence on full objective)**.** *Under Assumptions C.1–C.3, run stage $S$ for $T_S$ rounds with $K$ local steps, participation rate $q$, and $\eta_S \leq \frac{1}{4\bar{L}K}$. Then*

$$\frac{1}{T_S}\sum_{t=0}^{T_S-1}\mathbb{E}\|\nabla f(\Theta_t^{(S)})\|^2 \leq \frac{2\big(f(\Theta_0^{(S)}) - f_{\inf}\big)}{\eta_S K T_S} + \frac{\bar{L}\,\eta_S\,\sigma^2}{qN} + c_1\bar{L}^2\eta_S^2K^2G^2. \tag{24}$$

*Moreover, for any $\alpha > 0$,*

$$f(\widetilde{\Theta}_0^{(S)}) - f_{\inf} \leq \big(f(\Theta_0^{(S)}) - f_{\inf}\big) + \frac{1}{2\alpha}\|\nabla f(\Theta_0^{(S)})\|^2 + \frac{\alpha + L_f}{2}L(1 + \beta m_S)^2\big(\delta_{S-1}^{\mathrm{pre}}\big)^2. \tag{25}$$

*Proof.* Equation 24 is Theorem C.4 applied to stage $S$, where $F_S = f$. Equation 25 follows from Equation 23 with $\Theta = \Theta_0^{(S)}$, $\Delta = \widetilde{\Theta}_0^{(S)} - \Theta_0^{(S)}$, and Equation 22. □

**Corollary C.8** (Final-stage $O(1/\sqrt{T_S})$)**.** *If $\eta_S = \frac{c_0}{K\sqrt{T_S}}$ with $c_0 \leq \frac{1}{4\bar{L}}$, then*

$$\frac{1}{T_S}\sum_{t=0}^{T_S-1}\mathbb{E}\|\nabla f(\Theta_t^{(S)})\|^2 \leq O\bigg(\frac{f(\Theta_0^{(S)}) - f_{\inf}}{\sqrt{T_S}}\bigg) + O\bigg(\frac{1}{qN\,K\sqrt{T_S}}\bigg) + O\bigg(\frac{1}{T_S}\bigg). \tag{26}$$

*In addition, Equation 25 shows an explicit DBLF construction perturbation term $O\big(L(1 + \beta m_S)^2(\delta_{S-1}^{\mathrm{pre}})^2\big)$.*

**Conclusion.** The proof now matches the exact stage flow: regroup → DBLF construct submodel → FedAvg train → map back to full model, repeated across stages. Per-stage guarantees are established on the corresponding stage objectives, and the final guarantee is stated on a single full objective $f$, without telescoping across incompatible objectives.

# D MORE EXPERIMENTS

## D.1 GENERALIZABILITY ANALYSIS: CENTRALIZED TRAINING

We further evaluate the generalizability of DEVFT under centralized training. To this end, we conduct additional experiments using the same stage-wise training strategy, with end-to-end fine-tuning as the baseline. As shown in Table 8, DEVFT consistently outperforms end-to-end fine-tuning across all benchmarks. On LLaMA2-7B, DEVFT improves performance by 4.03%, 4.32%, 3.22%, and 2.71% on TruthfulQA, MMLU, IFEval, and BBH, respectively, while accelerating convergence by 3.22×. On the larger LLaMA2-13B, DEVFT maintains this advantage, yielding an average performance gain of 4.56% and a 3.04× speedup.

Table 9: Performance comparison under heterogeneous resource constraints on LLaMA2-7B (INT4).

| Method | Close-Ended Benchmark ↑ | | | | |
|---|---|---|---|---|---|
| | TruthfulQA | MMLU | IFEval | BBH | Average |
| FedIT | 41.96 | 40.82 | 26.38 | 37.13 | 36.57 (-2.87) |
| DoFIT | 43.74 | 41.53 | 28.71 | 38.45 | 38.11 (-1.33) |
| C2A | 41.78 | 40.63 | 26.21 | 36.98 | 36.40 (-3.04) |
| ProgFed | 45.04 | 41.67 | 29.26 | 38.51 | 38.62 (-0.82) |
| FLoRA | 43.58 | 41.35 | 28.06 | 37.94 | 37.73 (-1.71) |
| FedSA-LoRA | 45.27 | 41.77 | 29.60 | 38.62 | 38.82 (-0.62) |
| DEVFT | **45.86** | **42.35** | **30.54** | **39.01** | **39.44** |

These gains arise from DEVFT 's developmental training paradigm, which incrementally expands model capacity during fine-tuning. Progressive scaling smooths the loss landscape and mitigates poor local minima, while compact early-stage submodels reduce computation. The knowledge learned at each stage is then reused to initialize subsequent larger models, thereby accelerating convergence. Overall, these results demonstrate the generalizability of DEVFT beyond the federated setting, underscoring its potential as an efficient and scalable fine-tuning framework even under centralized training.

## D.2 ROBUSTNESS UNDER HETEROGENEOUS RESOURCE CONSTRAINTS

To evaluate the robustness of DEVFT in realistic edge environments, we simulate a heterogeneous setting where device memory budgets range from 3GB to 9GB. In this setup, resource-constrained devices (e.g., those with 3GB memory) participate exclusively in the early, less resource-intensive developmental stages. Experimental results in Table 9 demonstrate that DEVFT outperforms all baselines, achieving an average performance improvement of up to 3.04%. This performance boost is attributed to the inclusive nature of DEVFT: unlike baselines that exclude low-memory devices entirely, DEVFT enables these devices to contribute their valuable local data to the model's foundational knowledge during the early stages.

## D.3 COMPARISON WITH STATE-OF-THE-ART LORA OPTIMIZATION METHODS

To further validate the effectiveness of DEVFT, we conduct a comprehensive comparison against recent state-of-the-art LoRA optimization methods, including Fed-pilot (Zhang et al., 2024c), Fed-HeLLo (Zhang et al., 2025), FlexLoRA (Bai et al., 2024), and HETLoRA (Cho et al., 2024), using the LLaMA2-7B. As summarized in Table 10, DEVFT demonstrates superior performance across all evaluated benchmarks, achieving an average accuracy gain of up to 1.99% over baselines. We attribute this improvement to DEVFT's developmental training paradigm, which effectively navigates the optimization landscape to discover a superior convergence trajectory. It is important to note that these baselines primarily focus on optimizing LoRA module allocation or rank adaptation. In contrast, DEVFT targets the training process. Consequently, our method is not mutually exclusive but rather complementary to these approaches. DEVFT can potentially be combined with these LoRA-optimization techniques to further enhance federated fine-tuning performance.

## E DISCUSSION

**Analysis of Peak Memory Efficiency.** Peak GPU memory usage constitutes a pivotal constraint governing device eligibility in FL training. While DEVFT is primarily architected to optimize cumulative system efficiency, it concurrently alleviates peak memory burdens. As evidenced in Figure 7, peak memory usage in DEVFT's early phases is substantially mitigated—reduced by up to $4\times$—relative to baseline methods. Although peak memory in the final stage matches the baseline, the total resource savings over the entire training process are considerable. Furthermore, DEVFT remains orthogonal to specific peak-memory reduction techniques, such as Fed-pilot (Zhang et al.,

Table 10: Performance comparison with LoRA optimization methods on LLaMA2-7B (INT4).

| Method | Close-Ended Benchmark ↑ | | | | |
|---|---|---|---|---|---|
| | TruthfulQA | MMLU | IFEval | BBH | Average |
| Fed-pilot | 48.15 | 42.84 | 32.24 | 39.41 | 40.66 (-1.67) |
| Fed-HeLLo | 48.23 | 42.96 | 32.37 | 39.54 | 40.78 (-1.55) |
| FlexLoRA | 47.83 | 42.72 | 32.15 | 39.36 | 40.52 (-1.81) |
| HETLoRA | 47.71 | 42.58 | 31.96 | 39.12 | 40.34 (-1.99) |
| DEVFT | **50.28** | **44.15** | **33.97** | **40.93** | **42.33** |

2024c). Since each DEVFT stage adheres to standard FL protocols, these optimization methods can be seamlessly integrated to further lower hardware barriers.

**Adaptive Scheduling Strategy.** In this work, we employ a globally fixed scheduling strategy. This design choice is motivated by the need for a controlled evaluation of DEVFT 's core contributions—specifically the developmental training paradigm, deconfliction-guided layer grouping (DGLG), and differential-based layer fusion (DBLF)—ensuring both reproducibility and fair comparison against baselines. This approach aligns with established protocols in resource-efficient FL literature, such as ProgFed (Wang et al., 2022). Notably, even with this fixed schedule, DEVFT demonstrates significant performance gains, achieving up to $4.59\times$ faster convergence and $10.67\times$ greater communication efficiency (as illustrated in Figures 5 and 6). However, we recognize that adaptively scheduling each stage represents a promising avenue for further optimization. As a general and extensible framework, DEVFT readily supports the integration of such adaptive mechanisms, which we leave for future exploration.

**Distinction from Continual Learning.** While the concept of dynamic model capacity has been explored in Continual Learning (CL) (Hung et al., 2019), our objective differs fundamentally from this paradigm. The primary goal of CL is to mitigate catastrophic forgetting by expanding the model to accommodate a sequence of distinct tasks. In contrast, DEVFT employs developmental growth as a resource-efficient training strategy for a single downstream task within a constrained federated environment. Rather than serving as a mechanism to preserve knowledge across different tasks, our approach leverages dynamic capacity specifically to navigate hardware bottlenecks on edge devices.

## F  THE USE OF LARGE LANGUAGE MODELS

A large language model was used solely as a general-purpose tool for linguistic polishing (e.g., grammar, wording, and clarity). The LLM did not generate research ideas, design experiments, write substantive sections, produce or analyze data, or create code. All technical content, claims, and conclusions were authored and verified by the human authors, who take full responsibility for the manuscript. Suggested edits from the LLM were reviewed and post-edited to avoid plagiarism, inaccuracies, or fabricated statements. The LLM is not an author.

## G  LIMITATIONS

While our proposed DEVFT demonstrates superior performance, several limitations warrant acknowledgment. First, our current research primarily focuses on federated learning within a single organization. Extending our method to cross-organizational collaborative scenarios, where addressing incentive mechanisms, trust establishment, and privacy concerns becomes paramount, represents a significant yet valuable direction for future investigation. Second, although our approach substantially reduces computational requirements compared to traditional methods, the overall environmental footprint of training LLMs remains considerable. Future work should more comprehensively quantify the carbon emission reductions achieved through our developmental paradigm and explore additional algorithmic and system-level optimizations to further minimize environmental impact.

