# OpenReview forum: "Developmental Federated Tuning: A Cognitive-Inspired Paradigm for Efficient LLM Adaptation"
_ICLR.cc/2026/Conference — ICLR 2026 Poster_

### Official Review · Reviewer_wUou · 2025-10-16

**Soundness:** 3
**Presentation:** 2
**Contribution:** 2
**Rating:** 4
**Confidence:** 4

**Summary:**

This paper presents DEVFT, a federated fine-tuning framework designed to reduce the resource demands of large language model adaptation through cognitive developmental training. DEVFT progressively fine-tunes models across multiple developmental stages, each expanding parameter capacity. It employs a deconfliction-guided layer grouping mechanism and a differential-based layer fusion strategy to construct stage-specific submodels efficiently.

**Strengths:**

1.	The addressed problem (how to design stage-specific submodels that facilitate progressive knowledge transfer while optimizing overall performance) is both practical and relevant to real-world federated learning scenarios.

2.	The study is supported by comprehensive experimental evaluation across diverse benchmarks.

**Weaknesses:**

1.	The motivation could be strengthened. The framework assumes limited device resources only in the initial training phase, yet it remains unclear why smaller submodels are necessary if participating devices can accommodate full-model fine-tuning in the end. Furthermore, while the approach draws inspiration from human cognitive development, its applicability to heterogeneous resource settings warrants deeper justification—especially in cases where client resources evolve over time. I.e., for the cases that there are some devices who have more compute/memory resources available in the initial phase of the training and some clients who cannot accommodate larger models for the later half of the training.

2.	The connection between LoRA/PEFT's limitations and federated-specific constraints is not clearly articulated, raising the question of whether the proposed solution addresses a genuinely federated challenge or a more general fine-tuning issue.

3.	The layer grouping mechanism lacks empirical evidence for claims such as "opposite signs neutralizing each other’s unique contributions," and the conceptual clarity of this section could be improved.

4.	The assertion that “redundant layers limit representational diversity” conflicts with the subsequent need for a larger model, suggesting an inconsistency in the motivation for model expansion.

**Questions:**

1.	Does the framework assume identical or distinct data distributions across developmental stages, and how does this affect convergence?

2.	Is there reference to federated-specific parameter-efficient fine-tuning (pEFT) literature in Section 2.1, or are the cited works limited to centralized settings?

3.	Does each developmental stage correspond to a single global model update, or are multiple federated communication rounds performed within a stage?

4.	In the differential-based layer fusion process, how are non-linear interactions between layers preserved when performing linear subtraction-based fusion?

---

> ### Author Response · Authors · 2025-11-21
> **Response to Reviewer wUou (1 / 4)**
>
> We sincerely thank you for your insightful and constructive review. We have carefully considered each point you raised and have incorporated corresponding revisions and clarifications into the uploaded revised manuscript. Below are our point-by-point responses to your concerns.
>
>
>
> > **W1: Motivation & Heterogeneous Resources.**
>
> We thank the reviewer for this insightful comment.
>
> **1. Why Submodels are Necessary?** We distinguish between two critical bottlenecks that edge devices face:
>
> - **Peak Memory (Feasibility):** As shown in Figure 7, our early-stage submodels reduce peak memory usage by up to 4x. For strictly resource-constrained devices, this is the **"enabling factor"** that determines whether participation is physically possible.
> - **Computational Cost (Efficiency):** Even if memory permits, the high per-round TFLOPs of end-to-end fine-tuning cause prohibitive battery drain and thermal throttling. By leveraging lightweight submodels in early stages, DevFT significantly reduces the cumulative computational burden, resulting in drastic savings in total training time and energy for all devices.
>
> **2. Heterogeneous and Evolving Resources:** DevFT's staged architecture is inherently designed for the dynamic heterogeneity. It allows for flexible, resource-adaptive participation:
>
> - **High-Resource Clients:** Participate in all stages for maximum contribution.
> - **Evolving-Resource Clients:** Can dynamically join or drop out based on real-time status.
> - **Low-Resource Clients:** Instead of being excluded entirely, they can contribute valuable data to the foundational early stages.
>
> **3. Performance under Heterogeneity:** To empirically validate this, we simulate a heterogeneous environment with device memory capacity varying from 3GB to 9GB. Clients participate only in stages compatible with their memory. DevFT significantly outperforms all baselines (by up to 3.04%). This is because DevFT can inclusively utilize the data from these resource-constrained devices, which are completely excluded from baseline training. This inclusive participation is precisely what makes our method robust to real-world heterogeneity. Relevant experiments have been included in **Appendix D.2**.
>
> **Table: Performance comparison under heterogeneous resource constrains on LLaMA2-7B (INT4).**
>
> | **Method** | **TruthfulQA** | **MMLU** | **IFEval** | **BBH** | Average             |
> | ---------- | -------------- | -------- | ---------- | ------- | ------------------- |
> | FedIT      | 41.96%         | 40.82%   | 26.38%     | 37.13%  | __36.57% (-2.87%)__ |
> | DoFIT      | 43.74%         | 41.53%   | 28.71%     | 38.45%  | __38.11% (-1.33%)__ |
> | C2A        | 41.78%         | 40.63%   | 26.21%     | 36.98%  | __36.40% (-3.04%)__ |
> | ProgFed    | 45.04%         | 41.67%   | 29.26%     | 38.51%  | __38.62% (-0.82%)__ |
> | FLoRA      | 43.58%         | 41.35%   | 28.06%     | 37.94%  | __37.73% (-1.71%)__ |
> | FedSA-LoRA | 45.27%         | 41.77%   | 29.60%     | 38.62%  | __38.82% (-0.62%)__ |
> | **DevFT**  | 45.86%         | 42.35%   | 30.54%     | 39.01%  | __39.44%__          |
>
>
>
> > **W2: Connection Between PEFT Limitations and Federated-Specific Constraints.**
>
> We appreciate this constructive comment. You are correct that the principles of our developmental method are generalizable. However, our solution is specifically designed to address critical hardware bottlenecks inherent to the federated edge—bottlenecks that standard PEFT methods (e.g., LoRA) fail to resolve.
>
> **1. Why Standard PEFT is Not Enough:** PEFT reduces trainable parameters but retains the **full base model**, leaving two critical barriers:
>
> - **Peak Memory (Feasibility):** Clients are required to load the **entire base model** into memory. For LLMs, this static memory footprint often exceeds the capacity of edge devices, preventing participation. DevFT addresses this by constructing variable-depth submodels, allowing devices to participate according to their capacity.
> - **Computational Cost (Sustainability):** Even with frozen parameters, the per-round computational cost (TFLOPs) of a full forward/backward pass remains substantial (see Fig. 1). This makes on-device training impractically slow and energy-intensive.
>
> **2. Generalizability vs. Necessity:** While DevFT is indeed a generalizable paradigm, its role fundamentally shifts when applied to FL:
>
> - **Generalizability:** As shown in Appendix D.1 (Table 8), DevFT outperforms end-to-end tuning in centralized settings (3.22x speedup, 3.57% gain on LLaMA2-7B).
> - **Necessity (The "Enabling" Factor):** In FL, this efficiency gain transforms from a "beneficial optimization" (on servers) to an **"enabling necessity"** (on edge devices). The 10.3x reduction in training time and 4x reduction in peak memory (Fig. 7, early stages) are not just improvements—they are the deciding factors that allow resource-constrained devices to participate at all. Without DevFT, standard PEFT remains infeasible for these clients.

---

> ### Author Response · Authors · 2025-11-21
> **Response to Reviewer wUou (2 / 4)**
>
> > **W3: Empirical Evidence and Rationale for Layer Grouping.**
>
> We apologize for the ambiguity and appreciate the opportunity to clarify the mechanism behind our grouping strategy.
>
> **1. Conceptual Rationale:** The fundamental goal of our method is to fuse a group of layers ($\mathbf{g}_n$) into a single, high-fidelity representative layer ($\vartheta^{\mathbf{g}_n}$). The efficacy of this information compression relies heavily on similarity: fusing layers that are already structurally or functionally similar minimizes information loss. Conversely, fusing highly dissimilar layers results in a low-fidelity representation that fails to preserve unique layer functions. Therefore, the specific objective of our **DGLG (Section 3.2)** is to identify an optimal grouping configuration that **maximizes intra-group similarity** prior to fusion. This ensures that the resulting submodel maintains high representational fidelity. We have explicitly added this rationale to **Section 3.2**.
>
> **2. Empirical Evidence:** This design is not merely theoretical; it is rigorously substantiated by our ablation study in Table 2 (Lines 413-423). We benchmark DGLG against two similarity-agnostic baselines: 'RANDOM' (stochastic grouping) and 'EVEN' (naive uniform grouping). The results on LLaMA3.1-8B are striking: 'RANDOM' and 'EVEN' groupings cause massive performance drops of 3.56% and 6.49%, respectively. This empirically proves two points:
>
> - **Non-Triviality:** Layer grouping is a critical step; naive or random approaches cause severe degradation.
> - **Effectiveness:** Our similarity-driven DGLG strategy successfully clusters layers with high similarity, thereby facilitating the construction of superior submodels.
>
>
>
> > **W4: Inconsistency between Redundancy and Model Expansion.**
>
> We apologize for the confusion. We wish to clarify that these two statements are not contradictory; rather, they address two distinct scales of the learning problem: the **Macro-Objective (Model Capacity)** and the **Micro-Optimization (Fusion Fidelity)**.
>
> **1. Macro-Objective (Why we expand):** The "need for a larger model" refers to the **total task capacity**. A larger model (e.g., LLaMA2-13B) inherently possesses stronger representational capabilities than a smaller one (e.g., LLaMA2-7B), which is requisite for high performance on complex tasks. Our developmental paradigm is a method to efficiently train this high-capacity target model.
>
> **2. Micro-Optimization (Why we avoid redundancy):** The phrase "redundancy limits diversity" refers specifically to the **fidelity of layer fusion** when constructing submodels. To enable training on the edge, we build small proxy submodels by fusing groups of layers.
>
> - **The Pitfall:** Naive fusion (like our 'SUM' baseline in Table 3) simply sums similar layers. This creates a redundant representation that amplifies common signals while suppressing unique functional details.
> - **Our Solution (DBLF):** In contrast, DBLF extracts only the differential information ($\theta_j - \theta_{\text{anchor}}$) from each layer. This creates a representative layer with high information density and diversity, ensuring the submodel remains an effective proxy.
>
> Therefore, our goal is to obtain a high-capacity final model. To do this efficiently on the edge, we must build low-capacity submodels. To make these submodels effective, their fused layers must be informationally diverse (non-redundant). The significant performance drop of the 'SUM' baseline confirms this logic.

---

> ### Author Response · Authors · 2025-11-21
> **Response to Reviewer wUou (3 / 4)**
>
> > **Q1: Data Distributions Across Developmental Stages.**
>
> We apologize for any ambiguity. We wish to clarify that in our standard experimental setup (Appendix C), local data partitions are assigned at initialization and remain fixed throughout the process. The term "development" pertains strictly to the **model architecture's evolution**, not a curriculum of data; thus, the local data distribution is consistent across stages. However, to fully address your concern, we evaluate DevFT's robustness to changing data distributions in two specific ways.
>
> **1.Static Data Heterogeneity (Non-IID):** We conduct experiments on BERT and RoBERTa with 100 to 10,000 clients using Non-IID partitioning (Dirichlet $\alpha=1$). As shown in the tables below, DevFT consistently outperforms the baseline. We attribute this to the developmental paradigm: smaller submodels in early stages act as implicit regularization, effectively mitigating client drift and local overfitting. These results are included in **Section 4.6**.
>
> **Table: Performance comparison on BERT under non-IID settings (Dirichlet $\alpha=1$).**
>
> | Method    | YELP-P          | AGNEWS          | YAHOO           | 20NEWS          | Average             |
> | --------- | --------------- | --------------- | --------------- | --------------- | ------------------- |
> | FedIT     | 83.12%          | 87.05%          | 68.34%          | 76.89%          | 78.85%              |
> | **DevFT** | 84.53% (+1.41%) | 90.91% (+3.86%) | 70.67% (+2.33%) | 80.06% (+3.17%) | **81.54% (+2.69%)** |
>
> **Table: Performance comparison on RoBERTa under non-IID settings (Dirichlet $\alpha=1$).**
>
> | Method    | YELP-P          | AGNEWS          | YAHOO           | 20NEWS          | Average             |
> | --------- | --------------- | --------------- | --------------- | --------------- | ------------------- |
> | FedIT     | 82.93%          | 87.86%          | 68.21%          | 77.32%          | 79.08%              |
> | **DevFT** | 84.02% (+1.09%) | 90.27% (+2.41%) | 71.35% (+3.14%) | 79.83% (+2.51%) | **81.37% (+2.29%)** |
>
> **2. Dynamic Data Distribution (Evolving Client Pool):** We also test a scenario where the effective global data distribution shifts across stages. As detailed in our response to **W1**, we simulate heterogeneous resource constraints where low-memory clients participate in early stages but drop out of later ones. This causes the active data pool to change stage-by-stage. Even in this dynamic setting, DevFT demonstrates superior performance (up to 3.04% average gain). Relevant experiments have been included in **Appendix D.2**.
>
> These results, supported by our convergence proof (Appendix A), demonstrate that DevFT is robust to both static (Non-IID) and dynamic (evolving-pool) data distributions.
>
>
>
> > **Q2: Reference Literature in Section 2.1.**
>
> We appreciate this opportunity to clarify the scope of our related work. Section 2.1 is exclusively dedicated to PEFT literature within the FL domain. We apologize if this distinction was not explicitly stated. The references cited in this section are strictly chosen to reflect **federated-specific** implementations and challenges, categorized as follows:
>
> - **Federated Prompt Tuning:** [1], [2], [3]
> - **Federated Adapter Tuning:** [4], [5]
> - **Federated LoRA Tuning:** [6], [7], [8], [9], [11]
>
> We believe this specific focus is essential to correctly **contextualize our contribution** against relevant FL baselines rather than general centralized PEFT methods.

---

> ### Author Response · Authors · 2025-11-21
> **Response to Reviewer wUou (4 / 4)**
>
> > **Q3: Relationship between Developmental Stage and Communication Round.**
>
> We apologize for any ambiguity. Each developmental stage consists of multiple federated communication rounds. As outlined in Section 4.1 (Lines 310-312), the submodel capacity remains fixed throughout a stage, during which a standard FL process occurs repeatedly. The "global model update" (i.e., the knowledge transfer process in Step 3 of Fig. 3) **occurs exclusively** at inter-stage transitions, rather than at every round.
>
> **Example:** In our 400-round experiment for LLaMA2-13B, the process is divided into 4 stages. Each stage persists for 100 communication rounds. The Knowledge Transfer operation occurs only 3 times (between the 4 stages), ensuring efficiency. We have revised **Section 2.2 and Section 3.1** to explicitly articulate this hierarchical structure.
>
>
>
>
>
> > **Q4: Clarification on Non-Linearity in Layer Fusion.**
>
> This is a very keen and insightful question. We acknowledge that our Differential-Based Layer Fusion (DBLF) is, mathematically, a linear operation. However, we wish to clarify that DBLF is not intended to introduce new non-linearities, but rather to **preserve and modulate** the complex, intrinsic non-linear capabilities already encoded within the pre-trained parameters ($\theta_j$). Our approach functions as a **"Guided Parameter-Space Editing"** mechanism:
>
> 1. **Prerequisite (Guided Grouping):** First, DGLG (Section 3.2) ensures that the layers to be fused share high functional and parametric similarity.
> 2. **Mechanism (Guided Editing):** Leveraging this similarity, DBLF treats $\theta_{\text{anchor}}$ as a functional baseline and injects differential information ($\theta_j - \theta_{\text{anchor}}$) from neighboring layers. This acts as a fine-grained **modulation** of the anchor's existing non-linear manifold.
>
> **Empirical Validation:** Table 3 (Lines 413-423) supports this logic. DBLF significantly outperforms SUM and R-ONE. This demonstrates that while the fusion operator is linear, its application as a differential editor effectively creates a representative layer that retains the group's collective, non-linear representational power.
>
>
>
> ---
>
>
>
> Again, we thank the reviewer for the valuable feedback. Please let us know if there are any other questions or suggestions.
>
>
>
> **References:**
>
> [1] Promptfl: Let federated participants cooperatively learn prompts instead of models-federated learning in age of foundation
>
> model. 2023, TMC.
>
> [2] Efficient model personalization in federated learning via client-specific prompt generation. 2023, ICCV.
>
> [3] Federated adaptive prompt tuning for multi-domain collaborative learning. 2024, AAAI.
>
> [4] Fedadapter: Efficient federated learning for modern nlp. 2023, MobiCom.
>
> [5] Communication efficient federated learning for multilingual neural machine translation with adapter. 2023, Arxiv.
>
> [6] Selective aggregation for low-rank adaptation in federated learning. 2025, ICLR.
>
> [7] Flora: Federated fine-tuning large language models with heterogeneous low-rank adaptations. 2024, NeurIPS.
>
> [8] Improving lora in privacy-preserving federated learning. 2024, ICLR.
>
> [9] Heterogeneous lora for federated fine-tuning of on-device foundation models. 2024, EMNLP.
>
> [10] Federa: Efficient fine-tuning of language models in federated learning leveraging weight decomposition. 2024, Arxiv.
>
> [11]  Federated lora with sparse communication. 2024, Arxiv.

---

> > ### Comment · Reviewer_wUou · 2025-11-21
> >
> > Thank you to the authors for their detailed response.
> > ___
> > 1. While the authors emphasize early-stage memory and compute savings through smaller submodels, it still raises a conceptual inconsistency: if devices are ultimately expected to fine-tune or run the full large model in later stages, then peak memory and computational constraints do not truly disappear, they are merely deferred. In practice, devices that cannot sustain the full model's memory footprint or thermal load at the end of training would still be unable to participate meaningfully, regardless of early-stage savings. I believe that optimizing only the early stages provides limited practical benefit unless the method also ensures that the final, full-model stage remains feasible within the same device constraints. Otherwise, the approach assumes a device capability that contradicts the very resource limitations used to justify submodels in the first place.
> >
> > Overall, I do not understand why including the devices with little resources for the early phase of the training is important.
> >
> > 2. In the same vein, because the approach still requires clients capable of accommodating the full model in the later stages of training, it is unclear how the method meaningfully addresses any FL-specific challenges. If only devices that can eventually handle the full model are eligible participants, the core resource-constraint issues in federated learning remain fundamentally unaddressed.
> >
> > 3. While examining Figure 7, the leftmost plot on Training Time raises a question: how is the per-round training time reported to be at most around 10 seconds? And why does the training time of DevFT is lower than that of FedIT, even in the last rounds (Rounds 225 to 300)?
> >
> > 4. Are error bars or standard deviations available for the results reported in Table 1?
> >
> > 5. Regarding the authors’ response to “Q1: Data Distributions Across Developmental Stages,” doesn’t a Dirichlet split with $\alpha=1$ correspond to a more IID (or less skewed) distribution? Typically, Dirichlet $\alpha \rightarrow 0$ produces increasingly non-IID, highly skewed client datasets.

---

> > > ### Author Response · Authors · 2025-11-23
> > > **Response to Reviewer wUou's Follow-up Comments (1 / 2)**
> > >
> > > We sincerely thank the reviewer for the prompt and engaging follow-up.
> > >
> > > > **Q1 & Q2: The Fundamental Value of Early-Stage Participation (Addressing "Conceptual Inconsistency" & "Host Full-Model Assumption").**
> > >
> > > We thank the reviewer for this critical and insightful comment. We understand the concern: if a device cannot handle the final model, does its early participation truly matter? We respectfully clarify that directly excluding low-resource devices (standard FL) can significantly compromise model performance. In FL, each device holds unique and valuable local data. Excluding these low-resource devices prevents the global model from effectively utilizing their local data, thereby degrading the model's generalization capability. We present concrete evidence to demonstrate how DevFT resolves this resource constraint and why early-stage participation is essential for robust learning, even if those devices cannot complete the final stage.
> > >
> > > **1.Adaptation Failure in Standard FL**
> > >
> > > In a realistic heterogeneous setting (device memory ranging from 3GB to 9GB [1]), fine-tuning LLaMA2-7B (INT4) with standard settings (LoRA Rank 32, Batch Size 16, Max Seq Len 512) requires approximately ~8.19 GB of peak memory.
> > >
> > > - **The Participation Barrier:** In standard FL approaches (e.g., FedIT [2]), only devices with $>8.19$ GB memory can participate. This rigid threshold results in a participation rate of only ~15%, excluding 85% of the devices and their associated data.
> > >
> > > - **Performance Collapse:** This significant exclusion limits FedIT's performance to a mere 36.57%. Crucially, this is barely distinguishable from the model's **Zero-Shot Performance of 35.90%**. This indicates that standard FL fails to effectively adapt the global model to the downstream task because it cannot overcome resource limitations, thereby preventing the effective utilization of the local data available on these devices.
> > >
> > >
> > >
> > > **2.The "Relay" Solution: Breaking the Memory Wall via DevFT**
> > >
> > > DevFT fundamentally addresses this by decomposing the training into developmental stages (e.g., 4 $\rightarrow$ 8 $\rightarrow$ 16 $\rightarrow$ 32 layers):
> > >
> > > - **100% Initial Participation:** Even strictly constrained devices (e.g., 3GB memory) can effectively participate in the first stage (Submodel with 4 layers), contributing their valuable data to the global model's foundation.
> > > - **Permanent Knowledge Encoding:** Although these devices drop out as the model grows to 8 or 16 layers, their contributions are permanently encoded in the model's weights during the early consolidation phase. This is because the sub-models in subsequent stages inherit the knowledge acquired by the first-stage sub-model (Section 3.4).
> > > - **Refinement by High-Resource Peers:** High-resource clients then "take over the baton" to refine this robust, data-rich foundation in later stages.
> > >
> > >
> > >
> > > **3.Empirical Verification**
> > >
> > > The impact of this inclusive strategy is quantifiable. By allowing low-resource devices to contribute early (rather than excluding them entirely), DevFT achieves a performance of 39.44%, a significant **+2.87%** improvement over FedIT. This gain is directly attributable to the data contributed by devices participating in the early stages.
> > >
> > > **Conclusion:** Therefore, we respectfully clarify that there is no conceptual inconsistency in our approach. DevFT does not mandate that all devices participate in the final "Full-Model" stage. Standard FL assumes a device must host the full model to contribute data. This leaves the core FL resource challenge unaddressed. In contrast, DevFT enables low-resource devices to contribute data before they hit their resource limits. Our results prove that this "early-stage relay" is effective. By harvesting data from all devices in the early phases, DevFT constructs a high-performance global model that significantly outperforms the data-starved baseline. Thus, the early stages are not a "deferred constraint" or a wasted effort, but a **vital enabling mechanism** that solves the data availability problem caused by hardware constraints.

---

> > > ### Author Response · Authors · 2025-11-23
> > > **Response to Reviewer wUou's Follow-up Comments (2 / 2)**
> > >
> > > > **Q3: Clarification on Figure 7 (Training Time).**
> > >
> > > - **1. Why ~10 seconds per round?** This efficiency is a direct result of our high-end hardware specifications and experimental settings. As detailed in Appendix C (Lines 876-880), we utilize the NVIDIA H800 GPU (a high-end enterprise accelerator). Additionally, each round consists of only 10 local iterations (Lines 858-860) with a batch size of 16 using LoRA. To further maximize computational efficiency and maintain alignment with prior benchmarks (e.g., OpenFedLLM[3]), we adopt INT4 quantization for models (Line 877). Consequently, completing 10 iterations in ~10 seconds on an H800 is fully consistent with expected hardware performance.
> > >
> > > - **Why is DevFT faster than FedIT in the final stage?** Although the parameter count and model depth are identical in the final stage, the 1.44x speedup (Lines 407-410) stems from fundamental differences in implementation architecture:
> > >
> > >   - **FedIT (Standard Loading Overhead):** Baselines like FedIT typically load the pre-trained model via standard high-level APIs (e.g., Hugging Face's `AutoModel`). These generic model loaders often introduce implicit **system overheads**, such as complex inheritance wrappers, non-contiguous memory layouts from checkpoint loading, or redundant hooks designed for general-purpose inference/compatibility rather than pure training efficiency.
> > >
> > >   - **DevFT (Reconstruction-Based Efficiency):** In contrast, DevFT **explicitly reconstructs** the stage-specific submodel (even the full-sized one) by assembling the representative layers. This process effectively creates a "clean" and streamlined PyTorch model instance from scratch. By reconstructing the model layer-by-layer, we bypass the redundant structural overheads and memory fragmentation inherent in standard pre-trained model objects. Consequently, this reconstructed "clean" model executes the forward and backward passes more efficiently than the standard "heavy" object used by baselines.
> > >
> > >
> > >
> > > > **Q4: Error Bars or Standard Deviations for Table 1.**
> > >
> > > As stated in Appendix C (Lines 878-879), all experiments were repeated multiple times to ensure reliability, and the values reported in Table 1 represent the averaged results of these independent runs. The observed standard deviations are consistently low—ranging between $\pm 0.15$ for Close-ended benchmarks and $\pm 0.1$ for Open-ended benchmarks. These narrow intervals confirm that the performance gains of DevFT are **statistically significant and robust**, rather than artifacts of random variance. We will explicitly include these error bars in the final version of the manuscript to ensure completeness.
> > >
> > >
> > >
> > > > **Q5: Clarification on Dirichlet Distribution ($\alpha$).**
> > >
> > > We appreciate the opportunity to clarify our experimental setting and the choice of hyperparameters. Consistent with established benchmarks in FL (e.g., [4] [5]), the concentration parameter $\alpha$ of the Dirichlet distribution controls the degree of data heterogeneity. The spectrum is typically defined as follows:
> > >
> > > - $\alpha \to \infty$: Identical (IID) distribution.
> > > - $\alpha = 1.0$: Moderate to Strong Non-IID. In this setting, clients possess skewed label distributions while retaining sufficient diversity to represent realistic user behaviors. This is widely regarded as a standard benchmark for practical non-IID scenarios.
> > > - $\alpha = 0.1$: Extreme (Pathological) Non-IID. This setting creates highly partitioned data where clients may hold samples from only a single class.
> > >
> > > Therefore, our primary choice of $\alpha=1.0$ was selected to represent a challenging yet realistic federated environment, distinct from an IID setting. To further demonstrate the robustness of DevFT under extreme heterogeneity, we conduct an additional experiment using BERT with **$\alpha=0.1$**. As shown in the table below, even under this pathological skew, DevFT maintains its superiority, achieving an average performance improvement of 4.62% over FedIT.
> > >
> > > **Table: Performance comparison on BERT under extreme non-IID settings (Dirichlet $\alpha=0.1$).**
> > >
> > > | Method    | YELP-P | AGNEWS | YAHOO  | 20NEWS | Average             |
> > > | --------- | ------ | ------ | ------ | ------ | ------------------- |
> > > | FedIT     | 80.13% | 84.56% | 65.36% | 70.32% | 75.09%              |
> > > | **DevFT** | 83.76% | 88.83% | 68.91% | 77.35% | **79.71% (+4.62%)** |
> > >
> > >
> > >
> > > **References:**
> > >
> > > [1] FwdLLM: Efficient Federated Finetuning of Large Language Models with Perturbed Inferences. 2024 ATC.
> > >
> > > [2] Towards building the federatedgpt: Federated instruction tuning. 2024 ICASSP.
> > >
> > > [3] Openfedllm: Training large language models on decentralized private data via federated learning. 2024 KDD.
> > >
> > > [4] Feddc: Federated learning with non-iid data via local drift decoupling and correction. 2022 CVPR.
> > >
> > > [5] FedAdapter: Efficient Federated Learning for Modern NLP. 2023 MobiCom.

---

### Official Review · Reviewer_Sn7Y · 2025-10-30

**Soundness:** 3
**Presentation:** 3
**Contribution:** 2
**Rating:** 4
**Confidence:** 4

**Summary:**

This work addresses the resource overhead in federated fine-tuning for LLMs at the edge devices and proposes a method called developmental federated tuning. This method decomposes the fine-tuning process into multiple developmental stages, with each stage optimizing a sub-model with increasing parameter capacity. Experimental results show that the proposed method outperforms several existing baselines in terms of communication overhead and convergence speed.

**Strengths:**

+ This works aims at practical issues in federated instruction tuning on the edge.
+ Figures are organized in a good shape.

**Weaknesses:**

1. Lack of investigation on existing work: Gradually introducing knowledge has already been demonstrated multiple times in research related to continual learning, such as in [1] (although [1] adjusts the model's capacity, which is not exactly the same as the technical approach in the current work). Additionally, the manuscript highlights the communication efficiency of this method in the introduction but does not discuss some existing works that optimize communication efficiency, such as [2] and [3].
2. Lack of baseline: The currently referenced resource-aware methods are either not tailored for LLMs or have not been formally published. It is recommended to include highly relevant methods from top conferences or journals in the past two years as baselines.
3. The relationship between this developmental tuning method and FL is unclear. The current method design may yield certain effects at the edge and seems to remain valid even without the distributed training architecture of FL.
4. This work uses a scenario with only 20 clients, which is too few for the cross-device scenario targeted by this work. It is recommended to conduct evaluations in scenarios with a larger number of clients and greater data heterogeneity.
5. The paper lacks theoretical analysis of convergence and does not provide a comparison of convergence curves, raising concerns about the method's convergence performance.

[1] Compacting, Picking and Growing for Unforgetting Continual Learning. NeurIPS 2019.

[2] Federated full-parameter tuning of billion-sized language models with communication cost under 18 kilobytes. ICML 2024.

[3] FwdLLM: Efficient Federated Finetuning of Large Language Models with Perturbed Inferences. ATC 2024.

**Questions:**

1. The paper introduces its method design by drawing an analogy to human growth. However, the final method design mainly focuses on progressively adjusting the scale of trainable model parameters. In human growth, different stages may involve learning knowledge of varying complexity. Could the method be further optimized by perhaps adjusting the learning difficulty or task complexity at different stages?
2. Please refer to the Weaknesses.

---

> ### Author Response · Authors · 2025-11-21
> **Response to Reviewer Sn7Y (1 / 3)**
>
> We sincerely thank you for your insightful and constructive review. We have carefully considered each point you raised and have incorporated corresponding revisions and clarifications into the uploaded revised manuscript. Below are our point-by-point responses to your concerns.
>
>
>
> > **W1: Lack of Investigation on Existing Work.**
>
> We thank the reviewer for highlighting these important connections and providing valuable references. We offer the following clarifications regarding the distinction and compatibility of our work:
>
> - **Connection to Continual Learning[1]:** This is an insightful connection. We acknowledge the pioneering work in Continual Learning (CL) [1] regarding dynamic model capacity. However, we respectfully clarify that our objective is fundamentally distinct. The primary goal of CL is to mitigate catastrophic forgetting, allowing a model to grow to accommodate a sequence of distinct tasks. In contrast, DevFT focuses on resource efficiency for a single downstream adaptation task within constrained federated environments. In essence, our developmental growth is a training strategy designed to navigate hardware bottlenecks on edge devices, rather than a mechanism to preserve knowledge across differing task distributions. We have added a detailed discussion to **Appendix E**.
> - **Communication Efficiency[2] [3]:** We appreciate you highlighting these excellent works on communication-efficient FL. First, we wish to clarify that DevFT's core innovation lies in reducing **on-device computation overhead** via its developmental paradigm. The significant communication savings (Fig. 6) are a substantial and beneficial **byproduct** of this compute-centric strategy, rather than the sole optimization target. Furthermore, we view techniques in [2] [3] (e.g., zeroth-order optimization) as orthogonal and complementary to our approach. DevFT optimizes the training trajectory (i.e., what to train and when), whereas works like [2] [3] optimize the parameter exchange protocol (i.e., how to communicate). As noted in Section 4.6, each DevFT stage adheres to standard FL processes, meaning advanced protocols like FedKSeed [2] can be integrated within our stages to achieve **cumulative efficiency gains**. We have added a discussion to **Section 2.1**.
>
>
>
> > **W2: Lack of Strong, Relevant Resource-Aware Baselines.**
>
> Thank you for this constructive comment. We appreciate the opportunity to strengthen our evaluation.
>
> **1. Rationale for Original Baselines:**
>
> - **ProgFed (ICML 2022):** We choose this as it is a foundational progressive resource-aware method in the field of FL. Although not originally tailored for LLMs, its core idea of progressive training is conceptually relevant to our developmental paradigm, making it a suitable comparison.
> - **FLoRA (NeurIPS 2024) & FedSA-LoRA (ICLR 2025):** These two works are indeed recent SOTA methods published in top-tier conferences. They are highly relevant resource-aware approaches for federated LLM fine-tuning. We have updated the citations in the revision to reflect their latest publication status.
>
> **2. Comparison with Additional SOTA Methods:** We further conduct new experiments against three additional SOTA resource-aware frameworks: Fed-HeLLo (TNNLS 2025) [4], FlexLoRA (NeurIPS 2024) [5], and HETLoRA (EMNLP 2024) [6]. Results demonstrate that DevFT consistently outperforms these baselines across all benchmarks, achieving average performance gains of 1.55%, 1.81%, and 1.99%, respectively. This superiority stems from our developmental paradigm, which optimizes the whole training trajectory rather than just the adaptation parameters. Relevant experiments have been included in **Appendix D.5**.
>
> **3. Complementarity:** Crucially, while these methods primarily optimize the LoRA modules, DevFT optimizes the underlying training process. Consequently, our approach is **orthogonal and complementary** to these techniques. DevFT can be combined with them to unlock further efficiency gains.
>
> **Table: Performance comparison with resource-aware related work on LLaMA2-7B.**
>
> | **Method**   | **TruthfulQA** | **MMLU** | **IFEval** | **BBH** | Average         |
> | ------------ | -------------- | -------- | ---------- | ------- | --------------- |
> | Fed-HeLLo[4] | 48.23%         | 42.96%   | 32.37%     | 39.54%  | 40.78% (-1.55%) |
> | FlexLoRA[5]  | 47.83%         | 42.72%   | 32.15%     | 39.36%  | 40.52% (-1.81%) |
> | HETLoRA[6]   | 47.71%         | 42.58%   | 31.96%     | 39.12%  | 40.34% (-1.99%) |
> | **DevFT**    | 50.28%         | 44.15%   | 33.97%     | 40.93%  | __42.33%__      |

---

> ### Author Response · Authors · 2025-11-21
> **Response to Reviewer Sn7Y (2 / 3)**
>
> > **W3: Relationship between The Developmental Tuning Method and FL.**
>
> We appreciate this insightful observation. You are entirely correct that our developmental method represents a generalizable training paradigm. However, its impact varies fundamentally between settings:
>
> **1. Generalizability (Centralized Setting):** DevFT is indeed effective beyond FL. As demonstrated in our centralized experiments (Appendix D.1, Table 8), DevFT outperforms standard end-to-end tuning on LLaMA2-7B, achieving a 3.22x speedup and a 3.57% performance gain. In a centralized context with powerful servers, this serves as a **valuable efficiency optimization**, saving time and computational costs.
>
> **2. Critical Necessity (Federated Setting):** In the resource-constrained federated setting, however, this efficiency gain transforms from "beneficial" to **"critical" and "enabling."**
>
> - **The Distinction:** On edge devices, the 10.3x reduction in per-round training time and 4x reduction in peak memory (Figure 7, the early stages) is the **enabling factor**. It determines whether the device can participate in the training at all.
> - **Breaking Barriers:** In many real-world scenarios, standard end-to-end training is prohibitively slow or strictly impossible due to OOM (Out-Of-Memory) errors. By initiating training with minimal submodels, DevFT allows these otherwise-excluded devices to participate.
>
> Therefore, while DevFT is a generalizable method, its primary contribution lies in **lowering the barrier to entry** for edge devices, making federated fine-tuning feasible where it was previously impossible.
>
>
>
> > **W4: Scalability with Large Client Numbers and Robustness to Data Heterogeneity.**
>
> We appreciate this constructive feedback. We fully agree that demonstrating scalability across a large client base and robustness to data heterogeneity is essential for verifying the method's applicability in real-world cross-device scenarios.
>
> **1. Rationale for Current Setup:**
>
> We respectfully clarify that our choice of 20 clients adheres to the standard evaluation protocol established by OpenFedLLM (KDD 2024). Aligning with this benchmark ensures that our results are directly comparable and reproducible against current state-of-the-art baselines. Furthermore, regarding data heterogeneity, our experiments focus on instruction-tuning tasks which lack explicit class labels, rendering traditional Non-IID partitions (such as label skew) inapplicable. To ensure consistency with prior art, we strictly followed the data partitioning protocols defined in OpenFedLLM [1] and FLoRA [2] (NeurIPS 2024).
>
> **2. New Experiments: Large-Scale and Non-IID Scenarios:**
>
> To rigorously address your concerns, we conduct an extensive new analysis on text classification tasks, scaling the system from 100 up to 10,000 clients under Non-IID data partitioning (Dirichlet $\alpha = 1$​). Specifically, the setup includes 100 clients for 20NEWS, 1,000 for AGNEWS and YELP-P, and 10,000 clients for YAHOO, utilizing BERT and RoBERTa as global models. As the results demonstrate (see tables below), DevFT consistently outperforms the baseline across all datasets, validating its robustness in large-scale, cross-device scenarios with data heterogeneity. We attribute this strong performance to our developmental paradigm: by initiating training with smaller, less complex submodels, DevFT effectively mitigates client drift and local overfitting. We have integrated this extensive analysis and discussion into **Section 4.6**.
>
> **Table: Performance comparison on BERT under non-IID settings (Dirichlet $\alpha=1$).**
>
> | Method    | YELP-P          | AGNEWS          | YAHOO           | 20NEWS          | Average             |
> | --------- | --------------- | --------------- | --------------- | --------------- | ------------------- |
> | FedIT     | 83.12%          | 87.05%          | 68.34%          | 76.89%          | 78.85%              |
> | **DevFT** | 84.53% (+1.41%) | 90.91% (+3.86%) | 70.67% (+2.33%) | 80.06% (+3.17%) | **81.54% (+2.69%)** |
>
> **Table: Performance comparison on RoBERTa under non-IID settings (Dirichlet $\alpha=1$).**
>
> | Method    | YELP-P          | AGNEWS          | YAHOO           | 20NEWS          | Average             |
> | --------- | --------------- | --------------- | --------------- | --------------- | ------------------- |
> | FedIT     | 82.93%          | 87.86%          | 68.21%          | 77.32%          | 79.08%              |
> | **DevFT** | 84.02% (+1.09%) | 90.27% (+2.41%) | 71.35% (+3.14%) | 79.83% (+2.51%) | **81.37% (+2.29%)** |

---

> ### Author Response · Authors · 2025-11-21
> **Response to Reviewer Sn7Y (3 / 3)**
>
> > **W5:  Lack of Theoretical Analysis and Convergence Curves.**
>
> - **Theoretical Analysis:** We respectfully wish to clarify that we provide a detailed, three-page theoretical analysis in **Appendix A**. This section rigorously establishes the convergence guarantees for DevFT, modeling both intra-stage optimization (within a single stage) and inter-stage dynamics (across developmental transitions). Specifically, in Equation (17), we derive an overall convergence rate. This rate matches that of standard end-to-end FedAvg, theoretically validating the convergence properties and efficiency of our method.
> - **Convergence Curves:** We appreciate this constructive suggestion. We agree that "Accuracy vs. Wall-clock Time" curves provide a direct visualization of the optimization trajectory. Our current figures (Figs. 5 & 6) were designed to highlight convergence efficiency—specifically, comparing the total cumulative time and communication required to reach convergence. Similarly, Figure 7 details the per-round resource consumption. We will include the suggested line graphs in the Appendix of the final revision.
>
>
>
> > **Q1: Analogy to Human Growth (Adjusting Learning Difficulty).**
>
> We sincerely thank the reviewer for this insightful comment and for engaging deeply with our core analogy. You are absolutely correct. The analogy to human development can be viewed from two primary angles:
>
> 1. **Capacity Curriculum:** The physical development of the brain (e.g., expanding neural capacity), which enables the processing of complex information.
> 2. **Data Curriculum:** The pedagogical progression of knowledge (e.g., mastering arithmetic before algebra, or simple syntax before complex reasoning).
>
> **Rationale for Our Focus:** Our current work focuses primarily on the first dimension—**Capacity Growth**. We respectfully clarify that this choice is driven by the specific constraints of our problem setting. Our primary objective is to resolve the physical hardware bottlenecks inherent in federated fine-tuning on edge devices. In this context, the immediate prohibitive obstacle is not the difficulty of the data, but the resource overhead of on-device model training. Therefore, we utilize the developmental analogy to design a training strategy that explicitly navigates these hardware limitations.
>
> **Future Outlook:** We fully agree that integrating a "data curriculum" (aligned with model growth) is a **logical and natural extension** of our framework. One can envision an advanced paradigm where the initial stage processes simpler instructions, evolving to handle abstract tasks in later stages. This synergy of model-level and data-level curricula represents an exciting frontier for realizing the full potential of the cognitive metaphor. We have incorporated this valuable suggestion into **Section 5** as a promising direction for future research.
>
>
>
> ---
>
>
>
> Again, we thank the reviewer for the valuable feedback. Please let us know if there are any other questions or suggestions.
>
>
>
> **References:**
>
> [1] Compacting, Picking and Growing for Unforgetting Continual Learning. NeurIPS 2019.
>
> [2] Federated full-parameter tuning of billion-sized language models with communication cost under 18 kilobytes. ICML 2024.
>
> [3] FwdLLM: Efficient Federated Finetuning of Large Language Models with Perturbed Inferences. ATC 2024.
>
> [4] Fed-HeLLo: Efficient Federated Foundation Model Fine-Tuning with Heterogeneous LoRA Allocation. IEEE TNNLS 2025.
>
> [5] Federated fine-tuning of large language models under heterogeneous tasks and client resources. NeurIPS 2024.
>
> [6] Heterogeneous LoRA for Federated Fine-tuning of On-Device Foundation Models. EMNLP 2024.

---

> ### Author Response · Authors · 2025-11-27
>
> Dear Reviewer Sn7Y,
>
> As the discussion phase is entering its final stage, we want to kindly follow up to ensure that our response has adequately addressed your concerns.
>
> We truly value the time you have dedicated to reviewing our work. If there are any remaining questions or if further clarification is needed, please let us know—we are eager to engage in further discussion to improve the paper.
>
> Thank you again for your constructive feedback.
>
> Best regards,
>
> The Authors

---

### Official Review · Reviewer_4FbG · 2025-10-31

**Soundness:** 3
**Presentation:** 3
**Contribution:** 3
**Rating:** 6
**Confidence:** 4

**Summary:**

This paper introduces DEVFT, a framework designed for resource-efficient federated fine-tuning of LLMs. DEVFT decomposes training into progressive stages of increasing model capacity, starting from a compact submodel. The method relies on two novel components: (1) **Deconfliction-Guided Layer Grouping (DGLG)**, which uses spectral clustering to group layers based on parameter similarity, and (2) **Differential-Based Layer Fusion (DBLF)**, which creates a representative layer for each group by fusing an anchor layer with parameter differentials. Knowledge is transferred between stages via LoRA parameters.

**Strengths:**

- **Novelty:** Introducing a developmental training framework to federated LLM tuning is creative and aligns with cognitive learning principles.
- **Technical soundness:** The DGLG and DBLF modules are mathematically clear and empirically validated through detailed ablations.
- **Practical benefits:** The approach significantly reduces communication and compute costs while improving accuracy on standard instruction-tuning tasks.
- **Compatibility:** The method can be combined with existing frameworks such as FedIT and FedSA-LoRA, further improving efficiency.
- **Reproducibility:** The authors provide source code, facilitating replication.

**Weaknesses:**

+ **Non-IID simulation:** Experiments are performed on datasets without explicit modeling of data heterogeneity, which is a core challenge in FL.
+ **Resource accounting scope:** The reported communication overhead accounts only for uplink transmission of LoRA parameters, while downlink costs (e.g., transferring dense submodels between stages) and server-side overheads for clustering and layer fusion are not explicitly quantified.
+ **Theory scope:** The convergence proof inherits classical FedAvg assumptions but does not analyze the approximation bias introduced by representative-layer fusion.
+ **Hyperparameter sensitivity:** The method depends on several hyperparameters ($\beta$, number of stages, initial capacity), yet sensitivity analysis is missing.

**Questions:**

1. **Layer adjacency constraint:** In Appendix B, each group's layers appear contiguous in the original model, but the clustering algorithm does not inherently enforce adjacency. How is this achieved or post-processed in implementation?
2. **Fusion weighting:** When constructing representative layers, have the authors considered similarity-weighted averaging (e.g., weighted by inter-layer cosine similarity) instead of uniform $\beta$-weighting? Would this further improve fusion fidelity?

---

> ### Author Response · Authors · 2025-11-21
> **Response to Reviewer 4FbG (1 / 3)**
>
> We sincerely thank you for your insightful and constructive review. We have carefully considered each point you raised and have incorporated corresponding revisions and clarifications into the uploaded revised manuscript. Below are our point-by-point responses to your concerns.
>
>
>
> > **W1: Performance Under Data Heterogeneity.**
>
> Thank you for this constructive comment. We agree that data heterogeneity is a core challenge in FL. We would like to clarify our rationale for the current experimental setup and demonstrate DevFT's robustness with new experiments.
>
> 1. **Rationale for Current Setup:** Our experiments focus on instruction-tuning tasks using datasets without clear class labels, making traditional Non-IID partitions like label skew difficult. To ensure consistency, we follow the data partitioning protocol from the OpenFedLLM[1] (2024, KDD), also used in recent works like FLoRA[2] (2024, NeurIPS).
>
> 2. **Robustness to Non-IID Settings**: To evaluate DevFT's robustness against data heterogeneity, we further conduct additional experiments on text classification tasks utilizing a Dirichlet distribution ($\alpha = 1$) for data partitioning. As illustrated in the table below, DevFT consistently outperforms the baseline in non-IID settings, achieving an average performance gain of up to 2.69% on BERT. This resilience stems from our developmental training paradigm: by commencing training with compact submodels, DevFT effectively mitigates client drift and reduces overfitting on heterogeneous local data. Furthermore, as detailed in Section 4.6, DevFT is fully compatible with existing non-IID techniques (e.g., FedProx [3], FedDC [4]), as each developmental stage adheres to standard FL protocols. We have incorporated these results and discussions into **Section 4.6**.
>
>    **Table: Performance comparison on BERT under non-IID settings (Dirichlet $\alpha=1$).**
>
>    | Method    | YELP-P          | AGNEWS          | YAHOO           | 20NEWS          | Average             |
>    | --------- | --------------- | --------------- | --------------- | --------------- | ------------------- |
>    | FedIT     | 83.12%          | 87.05%          | 68.34%          | 76.89%          | 78.85%              |
>    | **DevFT** | 84.53% (+1.41%) | 90.91% (+3.86%) | 70.67% (+2.33%) | 80.06% (+3.17%) | __81.54% (+2.69%)__ |
>
>    **Table: Performance comparison on RoBERTa under non-IID settings (Dirichlet $\alpha=1$).**
>
>    | Method    | YELP-P          | AGNEWS          | YAHOO           | 20NEWS          | Average             |
>    | --------- | --------------- | --------------- | --------------- | --------------- | ------------------- |
>    | FedIT     | 82.93%          | 87.86%          | 68.21%          | 77.32%          | 79.08%              |
>    | **DevFT** | 84.02% (+1.09%) | 90.27% (+2.41%) | 71.35% (+3.14%) | 79.83% (+2.51%) | **81.37% (+2.29%)** |
>
>
>
> > **W2: Communication and Computation Overhead**.
>
> We apologize for the lack of clarity regarding the resource accounting.
>
> 1. **Communication Cost:** We apologize for any ambiguity. Our analysis comprehensively accounts for both uplink and downlink communication costs. Specifically, clients download the base model parameters **only once** during the initialization phase. At the onset of each new developmental stage $s$, the server transmits only lightweight **control instructions** (e.g., layer indices for grouping and DBLF fusion operations) rather than base model parameters. Clients then utilize these instructions to **locally construct** the $L_s$-layer submodel from their stored base model parameters. Since the base parameters remain frozen and are never modified in our LoRA-based framework, they never require re-transmission. Consequently, the communication cost is dominated solely by the lightweight LoRA parameters, preserving DevFT's communication efficiency. A detailed discussion is provided in **Appendix E**.
> 2. **Server-Side Computation Overhead:** The server-side computation for DGLG (Section 3.2) and DBLF (Section 3.3) is **minimal**. These operations are one-off per stage, not per-round. Moreover, the computational cost is negligible. For example, for a 32-layer model, DGLG runs on a tiny 32x32 similarity matrix, and DBLF involves only simple parameter arithmetic. Therefore, compared to the massive computation performed by all clients, this server-side cost is insignificant, especially given that servers in FL systems are typically well-resourced.

---

> ### Author Response · Authors · 2025-11-21
> **Response to Reviewer 4FbG (2 / 3)**
>
> > **W3: Approximation Bias Introduced by Representative-Layer Fusion.**
>
> We apologize for any ambiguity. We wish to clarify that our convergence proof in Appendix A explicitly incorporates and analyzes the approximation bias introduced by the fusion process. The specific analysis proceeds as follows:
>
> 1. **Quantifying the Bias:** In Lemma A.1, we formalize the approximation bias (the distance between the stage $s$ model and stage $s+1$ initialization) and show it scales as:
>
>    $$||\theta_{0}^{(s+1)}-\theta_{T}^{(s)}||=O(\beta~\delta_{s}),$$
>
>    where $\delta_{s} = \max_{j,k \in \mathrm{g_n}} ||\theta_{j}^{(s)} - \theta_{k}^{(s)}||$ is the maximum parameter divergence within a group.
>
> 2. **Connecting to Algorithm Design:** This bound $O(\beta~\delta_{s})$ directly justifies our DGLG mechanism. The goal of DGLG is to find a partition that minimizes inter-group cuts based on similarity. This is precisely designed to maximize intra-group similarity, thereby minimizing $\delta_{s}$ and controlling the approximation bias.
>
> 3. **Global Convergence**: We then integrate this bounded bias term into the global convergence analysis (Appendix A.4), resulting in the following bound:
>
>    $$F_{S}(\theta_{T}^{(S)})-F_{1}(\theta_{0}^{(1)})\le-\sum_{s=1}^{S}[\frac{\eta_{s}K_{s}}{2T_{s}}\sum_{t}||\nabla F_{s}||^{2}-L\eta_{s}^{2}K_{s}^{2}G^{2}-O(\beta~\delta_{s})].$$
>
>    By ensuring $\beta\delta_s = O(\varepsilon^2)$ (which is achieved by our DGLG's fine-grained grouping) and selecting appropriate step-sizes, we guarantee that the global model reaches an $\varepsilon$-stationary point, thereby establishing the theoretical efficiency of DevFT.
>
>
>
> > **W4: Hyperparameter Sensitivity Analysis.**
>
> Thank you for your constructive comment. We would like to respectfully clarify that a sensitivity analysis regarding the number of stages and initial capacity is presented in **Section 4.6**. We also recognize the missing analysis for the fusion weight $\beta$ and provide new experimental results for it below.
>
> **1. Analysis of Initial Capacity:** Table 5 (Lines 447-456) examines the impact of varying initial submodel capacities \{1, 2, 4, 8, 16, 32\} on LLaMA3.1-8B. We observe that capacity 4 yields the optimal performance. Deviating from this optimal point results in degradation: an overly small capacity (e.g., 1) leads to a 2.69% average performance drop, while an overly large capacity (e.g., 16) causes a 3.97% decline. This phenomenon is analogous to human learning, where starting too early (infancy) or too late (adulthood) can lead to suboptimal outcomes.
>
> **Table: Performance analysis of different initial submodel capacities on LLaMA3.1-8B.**
>
> | Initial Capacity        | 1      | 2      | 4          | 8      | 16     | 32     |
> | ----------------------- | ------ | ------ | ---------- | ------ | ------ | ------ |
> | **Average Performance** | 61.56% | 62.78% | **64.25%** | 62.33% | 60.28% | 58.09% |
>
> **2. Analysis of Number of Stages:** The number of developmental stages is inversely determined by the submodel growth rate. Table 6 (Lines 447-456) analyzes the impact of growth rates \{2, 4, 8\}. We observe that a growth rate of 2 is optimal. More abrupt capacity transitions (associated with higher growth rates) significantly degrade performance. Specifically, for LLaMA2-13B, increasing the growth rate to 8 results in a substantial average performance drop of 11.6%. This phenomenon mirrors natural learning processes, where steady, incremental development typically yields superior long-term outcomes compared to aggressive, rushed expansion.
>
> **Table: Performance analysis under varying submodel growth rates on LLaMA2-7B.**
>
> | Growth Rate             | 2          | 4               | 8               |
> | ----------------------- | ---------- | --------------- | --------------- |
> | **Average Performance** | **42.33%** | 39.80% (-2.53%) | 37.08% (-5.25%) |
>
>  **Table: Performance analysis under varying submodel growth rates on LLaMA2-13B.**
>
> | Growth Rate             | 2          | 4              | 8               |
> | ----------------------- | ---------- | -------------- | --------------- |
> | **Average Performance** | **52.77%** | 46.47% (-6.3%) | 41.17% (-11.6%) |
>
> **3. New Analysis: Sensitivity to $\beta$.** We conduct additional experiments on LLaMA2-7B to investigate the impact of varying $\beta$ values on model performance. The empirical results indicate that performance remains highly stable across a wide range of $\beta$ values, demonstrating that our DBLF is robust to this hyperparameter choice. We have included this new analysis and the corresponding discussion in **Appendix D.6**.
>
>  **Table: Sensitivity analysis across varying $\beta$ on LLaMA2-7B.**
>
> | $\beta$ Value           | 0.05            | 0.10       | 0.15            | 0.20            |
> | ----------------------- | --------------- | ---------- | --------------- | --------------- |
> | **Average Performance** | 42.21% (-0.12%) | **42.33%** | 42.18% (-0.15%) | 41.96% (-0.37%) |

---

> ### Author Response · Authors · 2025-11-21
> **Response to Reviewer 4FbG (3 / 3)**
>
> > **Q1: Layer Adjacency Constraint.**
>
> Thank you for this constructive comment.
>
> 1. **Clarification of Appendix B:** The example in Appendix B (e.g., [\{$\theta_{1}$,$\theta_{2}$,$\theta_{3}$\}, \{$\theta_{4}$,$\theta_{5}$,$\theta_{6}$\}]) was intended as a simplified, pedagogical illustration to intuitively explain the multi-stage grouping concept.
> 2. **DGLG Implementation:** Our actual DGLG implementation is based purely on parameter similarity (Eq. 1) and does not enforce any adjacency constraint.
> 3. **Empirical Observation:** We have observed that neighboring layers often show high similarity in both function and parameters, as information processing in transformers is hierarchical. As a result, they are frequently grouped together by the DGLG. This empirical trend explains why the simplified, contiguous example is a useful and intuitive way to represent the process.
>
> We have included a detailed clarification in **Appendix B**.
>
>
>
> > **Q2: Layer Fusion Weighting.**
>
> This is an excellent suggestion. The rationale behind our uniform weighting factor $\beta$ in Equation (5) is to capture the differential information ($\theta_j - \theta_{\text{anchor}}$) from all non-anchor layers **in an unbiased manner**, ensuring that every layer's unique contribution is integrated equally.
>
> To validate your insightful alternative, we compare our DBLF (Ours) against a "Similarity-Weighted" fusion strategy, where the contribution of each differential is weighted by its similarity to the anchor (i.e., $\text{sim}(\theta_j, \theta_{\text{anchor}})$). The results indicate that our approach maintains a slight but consistent advantage across all models. We attribute this to the fact that DBLF is specifically designed to harvest diverse features from varying layers. Conversely, a similarity-weighted mechanism risks inadvertently dampening the unique information from layers that are less similar to the anchor (which often contain the most distinct features), thereby causing a loss of fidelity through over-smoothing.
>
>  **Table: Performance analysis under different layer fusion strategies.**
>
> | Method              | LLaMA2-7B       | LLaMA3.1-8B     | LLaMA2-13B      |
> | ------------------- | --------------- | --------------- | --------------- |
> | Similarity-Weighted | 42.07% (-0.26%) | 63.86% (-0.39%) | 52.45% (-0.32%) |
> | Ours                | **42.33%**     | **64.25%**     | **52.77%**     |
>
> ---
>
>
>
> Again, we thank the reviewer for the valuable feedback. Please let us know if there are any other questions or suggestions.
>
>
>
> **References**:
>
> [1] Openfedllm: Training large language models on decentralized private data via federated learning. KDD 2024.
>
> [2] Flora: Federated fine-tuning large language models with heterogeneous low-rank adaptations. NeurIPS 2024.
>
> [3] Federated Optimization in Heterogeneous Networks. MLSys 2020.
>
> [4] Feddc: Federated learning with non-iid data via local drift decoupling and correction. CVPR 2022.

---

### Official Review · Reviewer_fj71 · 2025-10-31

**Soundness:** 2
**Presentation:** 2
**Contribution:** 2
**Rating:** 4
**Confidence:** 5

**Summary:**

This paper presents DEVFT, a federated fine-tuning framework that reduces LLM adaptation costs through cognitive developmental training by progressively expanding submodel capacities across stages. Leveraging deconfliction-guided layer grouping and differential-based layer fusion, DEVFT achieves efficient and effective fine-tuning across multiple benchmarks.

**Strengths:**

The paper’s main strength lies in its effective reduction of resource consumption while maintaining strong performance across benchmarks.

**Weaknesses:**

1. The intuition behind the use case of this developmental process is unclear. While the method seems to optimize overall efficiency, it lacks discussion on peak efficiency (e.g., throughput, maximum GPU memory limits), which is often more critical in practical federated learning scenarios. To improve fairness and applicability, results should also be evaluated under varying resource constraints.

2. The paper lacks a clear system-level formulation of the FL process. Specifically, the grouping of parameters across clients is not well explained. In Equation (5), the meaning of θ is ambiguous and should be explicitly defined.

3. The current setup appears to assume that all clients have devices capable of hosting the full model. Can the method be extended to cases where not all layers are trainable due to device limitations?

4. It is also unclear how the method adapts to heterogeneous resources across clients. What strategies are employed for aggregation when clients train different subsets of layers?

5. Please consider including loss and accuracy curves. I am uncertain about the number of local and global training rounds, as downstream fine-tuning of LLMs typically requires fewer rounds. Further justification of the experimental settings would be helpful.

6. The choice of LoRA rank (e.g., 32) seems arbitrary. Since the optimal rank often depends on the downstream task, a fixed configuration might not generalize well. An ablation or explanation would strengthen this aspect.

7. The anchor layer is set to the first layer by default. Is there a particular reason for this choice? Would selecting a different anchor (e.g., middle or final layers) affect the performance?

8. The training rounds for each stage are fixed. Inspired by cognitive paradigms, one might argue that different stages (akin to learning phases) should have adaptive durations. For instance, adults may learn certain tasks faster than children and vice versa. Is the stage duration setting task-specific or task-agnostic?

9. If “capacity” refers to the number of unfrozen layers, what does the stage-wise scaling actually control? For example, the assumption of linear memory increase may not hold. As shown in Fed-Pilot [1], memory cost tends to grow non-linearly due to activation reuse and other factors. The effect of the scaling setting on the resource is worth discussing.

10. The implementation details of FedSA-LoRA as a baseline are unclear. In their original work, only the A matrix is shared across clients, and the server lacks a complete B matrix for global evaluation. How is this adapted in your framework for comparison?

11. Important prior works are missing from the related work section. For example, Fed-Pilot [1] provides a memory-aware LoRA allocation strategy; Fed-HeLLo [2] introduces heuristic and Fisher Information-based layer selection; FlexLoRA [3] and HETLoRA [4] address task and resource heterogeneity. These works should be discussed and compared if necessary.

12. Any generalizable findings from your experiments on the layer partition?

13. Lastly, Equation (5) appears to be a straightforward weighted average, as in standard FedAvg.

[1] Fed-pilot: Optimizing LoRA Allocation for Efficient Federated Fine-Tuning with Heterogeneous Clients. ArXiv 2024.

[2] Fed-HeLLo: Efficient Federated Foundation Model Fine-Tuning with Heterogeneous LoRA Allocation. IEEE TNNLS 2025.

[3] Federated fine-tuning of large language models under heterogeneous tasks and client resources. NeurIPS 2024.

[4] Heterogeneous LoRA for Federated Fine-tuning of On-Device Foundation Models. EMNLP 2024.

**Questions:**

See Weakness.

---

> ### Author Response · Authors · 2025-11-21
> **Response to Reviewer fj71 (1 / 4)**
>
> We sincerely thank you for your insightful and constructive review. We have carefully considered each point you raised and have incorporated corresponding revisions and clarifications into the uploaded revised manuscript. Below are our point-by-point responses to your concerns.
>
>
>
> > **W1: Intuition of DevFT & Peak Efficiency & Evaluation under Varying Resource Constraints.**
>
> Thanks for your constructive comment.
>
> - **Intuition:** Drawing inspiration from the progressive nature of human cognitive development, we aim to cultivate a robust LLM from a compact foundation, specifically tailored for resource-constrained federated fine-tuning. Given the strict computational and bandwidth limitations of edge devices, our primary objective is to optimize total resource consumption. By strategically utilizing computationally minimal submodels in the early stages, DevFT significantly reduces the resource overhead compared to conventional end-to-end training paradigms.
> - **Peak Efficiency:** We agree that peak efficiency (e.g., peak GPU memory) is a crucial metric because it determines a device's eligibility for training. We wish to clarify that while DevFT offers benefits in this area, its primary objective is to improve the overall efficiency of the entire federated system. As shown in Figure 7, DevFT's peak memory per stage is significantly lower (up to 4x) than baselines during the early stages. Although peak memory in the final stage matches the baseline, the total resource savings over the entire training process are considerable. Additionally, our method is **compatible with existing techniques aimed at reducing peak memory** (e.g., Fed-pilot[1]). This is because each stage in DevFT follows a standard FL process, allowing these techniques to be used directly. We have included a detailed discussion in **Appendix E**.
> - **Varying Resource Constraints:** We follow the reviewer's advice to test DevFT's robustness under different resource constraints. We simulate devices with 3-9GB memory, where low-memory devices can only participate in early, less demanding stages. As shown in the table below, DevFT outperforms all baselines, with up to 3.04% average performance improvement. This boost is due to DevFT's inclusive nature: resource-constrained devices, which would be excluded from baseline training, can still contribute to the early developmental stages. Relevant experimental results are included in **Appendix D.2**.
>
> **Table: Performance comparison under varying resource constrains on LLaMA2-7B (INT4).**
>
> | **Method** | **TruthfulQA** | **MMLU** | **IFEval** | **BBH** | Average             |
> | ---------- | -------------- | -------- | ---------- | ------- | ------------------- |
> | FedIT      | 41.96%         | 40.82%   | 26.38%     | 37.13%  | __36.57% (-2.87%)__ |
> | DoFIT      | 43.74%         | 41.53%   | 28.71%     | 38.45%  | __38.11% (-1.33%)__ |
> | C2A        | 41.78%         | 40.63%   | 26.21%     | 36.98%  | __36.40% (-3.04%)__ |
> | ProgFed    | 45.04%         | 41.67%   | 29.26%     | 38.51%  | __38.62% (-0.82%)__ |
> | FLoRA      | 43.58%         | 41.35%   | 28.06%     | 37.94%  | __37.73% (-1.71%)__ |
> | FedSA-LoRA | 45.27%         | 41.77%   | 29.60%     | 38.62%  | __38.82% (-0.62%)__ |
> | **DevFT**  | 45.86%         | 42.35%   | 30.54%     | 39.01%  | __39.44%__          |
>
>
>
> > **W2: FL System Formulation & Clarification of Equation (5).**
>
> - **FL System Formulation:** Sorry for the confusion. Our federated system has three steps (Figure 3):
>
>   1. **Stage Submodel Construction:** Prior to the commencement of each stage, the server constructs a stage-specific submodel. Specifically, the server utilizes the deconfliction-guided layer grouping (DGLG) mechanism to cluster layers exhibiting minimal parameter conflicts. Subsequently, the differential-based layer fusion (DBLF) strategy is applied to integrate intra-group information, generating a representative layer for each group. Finally, these representative layers are concatenated sequentially to assemble the stage-specific submodel, which is uniform across all devices.
>   2. **Collaborative Optimization:** Once the submodel is constructed, the federated fine-tuning process commences, wherein devices collaboratively train the submodel on their local data.
>   3. **Knowledge Transfer:** Upon completion of the current stage, the acquired knowledge is synchronized to update the global model and is seamlessly transferred to initialize the submodel for the subsequent stage. This progressive model training process continues until the completion of the S-th stage.
>
>   We have included a detailed description in **Section 3.1**.
>
> - **Meaning of** $\theta$​ **in Eq (5):** We apologize for the ambiguous notation. We have revised the paper (Line 254) to include the following precise definitions:
>
>   - $\theta_j$: Represents the parameters of the $j$-th layer.
>
>   - $\theta_{\text{anchor}}$: Represents the parameters of the anchor layer.

---

> ### Author Response · Authors · 2025-11-21
> **Response to Reviewer fj71 (2 / 4)**
>
> > **W3: Extension to Device Heterogeneity.**
>
> We thank the reviewer for this crucial point. Our framework is **extensible to device heterogeneity**, including scenarios where not all devices can host the full model. They can also contribute to the global model during early, low-resource stages (e.g., Stages 1 & 2), helping build robust foundational representations. This is validated by new experiments in **Response to W1**. Many low-memory devices (~3GB) can't participate in full-model training. However, DevFT effectively utilizes the data from these otherwise-excluded devices during its early stages, enabling it to achieve superior performance.
>
>
> > **W4: Handling Heterogeneous Resources & Aggregation Strategy.**
>
> Thanks for your constructive comment.
>
> - **Aggregation Strategy:** In our framework, all clients participating in a given stage $s$ train the **identical** submodel. Consequently, aggregation is handled using FedAvg.
> - **Heterogeneous Resources:** Our framework naturally adapts to system heterogeneity, as shown in **Response to W1**.
>
>
>
> > **W5: Convergence Curves & Training Rounds.**
>
> - **Convergence Curves:** Thank you for the suggestion. We will add detailed convergence curves (e.g., accuracy vs. time and loss vs. time) to the Appendix to provide a more complete picture of the training dynamics.
>
> - **Training Rounds:** We apologize for any lack of clarity. As detailed in **Appendix C**, we set a total of 300 global rounds for 7B/8B models and 400 rounds for the 13B model, with 10 local update steps per round. This configuration was deliberately chosen to ensure that all baselines achieve full convergence. It is also consistent with established protocols in OpenFedLLM [5] (KDD, 2024).
>
>
>
> > **W6: Choice of LoRA Rank.**
>
> Thank you for your constructive comment. We choose $r=32$ because it's a robust, common setting in federated LoRA fine-tuning (e.g., OpenFedLLM[5], FLoRA[6]). Our goal is to ensure a fair comparison against baselines under a standard configuration, thereby demonstrating DevFT's relative advantage, rather than to optimize the rank $r$ for this specific task.
>
> We acknowledge that the optimal rank $r$ is task-dependent. We conduct additional experiments on LLaMA2-7B across a spectrum of rank values. As shown in the table below, DevFT's performance improves consistently as the rank increases, demonstrating robust scalability. This consistency confirms that our main results at $r=32$ are representative and not an artifact of a specific hyperparameter choice. Detailed results have been included in **Appendix D.3**.
>
> **Table: Sensitivity analysis across different LoRA ranks on LLaMA2-7B (INT4).**
>
> | **Rank** | **TruthfulQA** | **MMLU** | **IFEval** | **BBH** | Average             |
> | -------- | -------------- | -------- | ---------- | ------- | ------------------- |
> | 16       | 50.16%         | 43.98%   | 33.75%     | 40.76%  | __42.16% (-0.17%)__ |
> | 32       | 50.28%         | 44.15%   | 33.97%     | 40.93%  | __42.33%__          |
> | 64       | 50.42%         | 44.37%   | 34.19%     | 41.21%  | __42.55% (+0.22%)__ |
>
>
>
> > **W7: Choice of the Anchor Layer.**
>
> We appreciate this comment. Our choice of the first layer as the anchor stems from our DGLG, which clusters layers with high parameter similarity. Due to this similarity, the anchor choice (first, middle, or last) minimally impacts performance. To validate this, we conduct a new ablation study. The performance difference is indeed negligible, which confirms the robustness of our DGLG and DBLF. While the differences are minor, the **First Layer** strategy consistently achieves the optimal performance. We attribute this to the fact that using the initial layer as the reference basis allows for a **more precise integration of the relative information gains**  provided by the subsequent layers within the cluster. Relevant experiments have been included in **Appendix D.4**.
>
> **Table: Performance comparison with different anchor layer selections on LLaMA2-7B (INT4).**
>
> | **Anchor Layer** | **TruthfulQA** | **MMLU** | **IFEval** | **BBH** | Average              |
> | ---------------- | -------------- | -------- | ---------- | ------- | -------------------- |
> | First Layer      | 50.28%         | 44.15%   | 33.97%     | 40.93%  | __42.33%__           |
> | Middle Layer     | 50.07%         | 43.94%   | 33.71%     | 40.79%  | **42.13% (-0.20%)** |
> | Last Layer       | 49.84%         | 43.85%   | 33.37%     | 40.52%  | **41.90% (-0.43%)** |

---

> ### Author Response · Authors · 2025-11-21
> **Response to Reviewer fj71 (3 / 4)**
>
> > **W8: Adaptive Scheduling Strategy**.
>
> We sincerely appreciate this insightful feedback. We agree that adaptively scheduling each stage is a very promising direction. However, our current design is motivated by the following considerations:
>
> 1) A globally fixed schedule enables controlled evaluation of DevFT’s core components—developmental training paradigm, deconfliction-guided layer grouping, and differential-based layer fusion—ensuring both reproducibility and fair comparison.
> 2) This design aligns with prior work such as ProgFed [7], adhering to established evaluation protocols in resource-efficient FL.
> 3) As shown in Figures 5 and 6, even with a fixed schedule, DevFT delivers significant improvements in convergence speed (up to 4.59×) and communication efficiency (up to 10.67×).
> 4) We fully agree that incorporating adaptive scheduling is a valuable direction. As discussed in Section 4.6, DevFT is a **general and extensible framework** that readily supports such enhancements in future work.
>
>  We have included a detailed discussion in **Appendix E**.
>
>
>
> > **W9: Definition of Capacity & Non-Linear Memory Scaling.**
>
> - **Definition of Capacity:** We apologize for any ambiguity. "Capacity," as defined in Section 2.2 (Lines 147-150), refers specifically to the number of layers of the submodel used in each developmental stage. For instance, for LLaMA2-7B (which has 32 total layers), the submodels in its four developmental stages consist of 4, 8, 16, and 32 layers, respectively.
> - **Non-Linear Memory Scaling:** We agree that the total training memory footprint comprises multiple components. As characterized in established works like Fed-pilot [1], memory usage consists of **foundational memory** (model parameters, optimizer states, and activations) and **system overhead** (e.g., CUDA context). In this paper, we focus on optimizing the foundational memory, as this is the dominant bottleneck and represents the theoretical lower bound of memory required for model deployment. Since system overhead is largely device-dependent and variable, Figure 7 specifically illustrates the scaling of this foundational memory, which scales approximately linearly with the number of layers. We have revised the **caption of Figure 7** to explicitly state that it reports the foundational memory footprint to avoid confusion.
>
>
>
> > **W10: Implementation of the FedSA-LoRA.**
>
> To ensure an accurate comparison, we follow the FedSA-LoRA setup using its **official open-source implementation**. Clients train both LoRA matrices locally but only upload A for aggregation. Applying FedSA-LoRA to our method requires no adjustments because each DevFT stage operates as a **standard FL process**. Thus, FedSA-LoRA can be directly applied at each stage of DevFT. This further shows DevFT is a general framework adaptable to other methods (as detailed in Section 4.6).
>
>
>
> > **W11: Missing Related Work.**
>
> Thank you for pointing out these highly relevant and excellent works [1-4]. We view our work as complementary to these methods, as they address different (yet equally important) aspects of the FL challenge:
>
> - **Fed-Pilot, et al. [1-4]** primarily focus on LoRA optimization—that is, how to adapt LoRA modules (e.g., rank or layers) for heterogeneous clients.
> - **DevFT**, in contrast, focuses on optimizing the training process itself (i.e., how to train) via a developmental paradigm. This means the optimization techniques from [1-4] can be applied within each DevFT stage to further enhance performance.
>
> Furthermore, to validate DevFT's effectiveness, we conduct additional experiments on LLaMA2-7B. The results underscore its strong performance, achieving an average gain of up to 1.99%. We attribute this improvement to DevFT's developmental training paradigm, which effectively navigates the optimization landscape to discover a superior convergence trajectory. We want to emphasize that these methods each offer unique benefits, and DevFT is compatible with them. Combining these could further advance federated fine-tuning. We have incorporated this discussion into **Section 2.1** and detailed the experimental results in **Appendix D.5**.
>
> **Table: Performance comparison with state-of-the-art LoRA optimization methods on LLaMA2-7B.**
>
> | **Method**   | **TruthfulQA** | **MMLU** | **IFEval** | **BBH** | Average         |
> | ------------ | -------------- | -------- | ---------- | ------- | --------------- |
> | Fed-pilot[1] | 48.15%         | 42.84%   | 32.24%     | 39.41%  | 40.66% (-1.67%) |
> | Fed-HeLLo[2] | 48.23%         | 42.96%   | 32.37%     | 39.54%  | 40.78% (-1.55%) |
> | FlexLoRA[3]  | 47.83%         | 42.72%   | 32.15%     | 39.36%  | 40.52% (-1.81%) |
> | HETLoRA[4]   | 47.71%         | 42.58%   | 31.96%     | 39.12%  | 40.34% (-1.99%) |
> | **DevFT**    | 50.28%         | 44.15%   | 33.97%     | 40.93%  | __42.33%__      |

---

> ### Author Response · Authors · 2025-11-21
> **Response to Reviewer fj71 (4 / 4)**
>
> > **W12: Generalizable Findings from Layer Partitioning.**
>
> Thank you for your constructive comment. We observe several generalizable findings from our layer partitioning analysis.
>
> - **Dynamic Grouping Patterns:**
>   - In the **early stages**, shallow (input) and deep (output) layers are often separated, indicating they contain more unique information. Intermediate layers, with more uniform transformations, tend to cluster together.
>   - In the **later stages**, this pattern evolves. The intermediate layers often begin to diversify into different groups, while the shallow and deep layers start to align and group together.
> - **Layer Importance:** We also assessed layer-wise importance. The first and last layers are highly important due to their roles in input transformation and output shaping.
>
> We have added a detailed discussion of these findings to the **Appendix E**.
>
>
>
> > **W13: Clarification on Equation (5).**
>
> We would like to clarify that Equation (5) is not a parameter aggregation formula. Instead, it is a core component of our submodel construction process.
>
> - **When/Where:** This operation happens on the server before a new developmental stage's training begins.
> - **Purpose:** Its purpose is to create a single "representative layer" by fusing a group of existing layers from the global model.
>
>
>
> ---
>
>
>
> Again, we thank the reviewer for the valuable feedback. Please let us know if there are any other questions or suggestions.
>
>
>
> **References**:
>
> [1] Fed-pilot: Optimizing LoRA Allocation for Efficient Federated Fine-Tuning with Heterogeneous Clients. ArXiv 2024.
>
> [2] Fed-HeLLo: Efficient Federated Foundation Model Fine-Tuning with Heterogeneous LoRA Allocation. IEEE TNNLS 2025.
>
> [3] Federated fine-tuning of large language models under heterogeneous tasks and client resources. NeurIPS 2024.
>
> [4] Heterogeneous LoRA for Federated Fine-tuning of On-Device Foundation Models. EMNLP 2024.
>
> [5] Openfedllm: Training large language models on decentralized private data via federated learning. KDD 2024.
>
> [6] Flora: Federated fine-tuning large language models with heterogeneous low-rank adaptations. NeurIPS 2024.
>
> [7] ProgFed: Effective, communication, and computation efficient federated learning by progressive training. ICML 2022.

---

> ### Author Response · Authors · 2025-11-27
>
> Dear Reviewer fj71,
>
> As the discussion phase is entering its final stage, we want to kindly follow up to ensure that our response has adequately addressed your concerns.
>
> We truly value the time you have dedicated to reviewing our work. If there are any remaining questions or if further clarification is needed, please let us know—we are eager to engage in further discussion to improve the paper.
>
> Thank you again for your constructive feedback.
>
> Best regards,
>
> The Authors

---

### Author Response · Authors · 2025-11-30
**TL;DR A Summary of Discussion by Authors**

**Dear Reviewers, AC, and Researchers,**

**Thank you all for your dedicated effort at this difficult time to our ICLR community.** We are grateful for the constructive feedback from four reviewers. In the revised PDF, we have incorporated substantial new experiments and clarifications in Section 3, Section 4.6, Section 5, and Appendices B-E to effectively address the reviewers' concerns.

We first recap the major concerns raised by reviewers and our corresponding solutions:

- **[Motivation & Robustness]:** Addressed concerns regarding the conceptual consistency, peak memory efficiency, and robustness under heterogeneous resource constraints. (W1&3&4@fj71,W1@wUou)
- **[Baselines]:** Addressed the need for comparisons against more resource-aware baselines. (W11@fj71, W1&2@Sn7Y)
- **[Scalability & Data Heterogeneity]:** Addressed concerns regarding performance on Non-IID data and scalability to large client populations. (W1@4FbG, W4@Sn7Y, Q1@wUou)
- **[Theoretical & Sensitivity Analysis]:** Addressed concerns regarding convergence guarantees, approximation bias, and hyperparameter sensitivity. (W6&7@fj71, W3&4@4FbG, W5@Sn7Y)

Now we summarize our discussion and the specific actions taken for each reviewer below:

**Reviewer fj71**

- **W1&3&4:** We conducted new experiments to demonstrate DevFT's robustness under varying resource constraints (Appendix D.2) and clarified our method's adaptability to peak memory constraints (Appendix E) and aggregation strategies.
- **W2&9&13:** We clarified the system workflow (Section 3.1) and explicitly defined model capacity alongside the notations in Eq. 5. We further clarified that Eq. 5 governs submodel construction rather than aggregation, and provided a detailed discussion on memory scaling (Fig. 7).
- **W5&10:** We clarified the training rounds (Appendix C) and the implementation details of FedSA-LoRA.
- **W6&7:** We performed sensitivity analyses on LoRA rank (Appendix D.3) and Anchor Layer selection (Appendix D.4).
- **W8&12:** We expanded the discussion in Appendix E to cover generalizable insights derived from layer partitioning and highlighted adaptive scheduling as a promising future direction.
- **W11:** We added comparisons against LoRA optimization methods (Appendix D.5) to address the concern of missing baselines.

**Reviewer 4FbG**

- **W1:** We conducted experiments on text classification under Non-IID settings (Section 4.6), proving DevFT's robustness to data heterogeneity.
- **W2:** We detailed the communication costs in Appendix E and explained that server-side computation is negligible.
- **W3:** We explicitly analyzed the approximation bias introduced by layer fusion in Appendix A.
- **W4:** We added a sensitivity analysis for the fusion weight (Appendix D.6) and validated the impact of submodel initial capacity and growth rates in Section 4.6.
- **Q1:** We clarified the implementation details in Appendix B to address the concern of layer adjacency constraints.
- **Q2:** We conducted additional experiments against a Similarity-Weighted fusion strategy, validating the superiority of our unbiased DBLF approach.

**Reviewer Sn7Y**

- **W1&2:** We incorporated a detailed discussion on continual learning (Appendix E), expanded Section 2.1 to contextualize communication-efficient works, and added comparisons against resource-aware baselines in Appendix D.5.
- **W3:** We clarified the critical necessity of DevFT for enabling edge participation in federated settings.
- **W4:** We conducted experiments with client scales ranging from 100 to 10,000 under Non-IID settings (Section 4.6), validating DevFT's scalability regarding client population and robustness to data heterogeneity.
- **W5&Q1:** We highlighted our theoretical convergence analysis (Appendix A) and discussed the potential of integrating a data curriculum in Section 5.

**Reviewer wUou**

- **W1:** We addressed the conceptual inconsistency concern by detailing the "Relay" mechanism and empirically proved that including low-resource devices boosts model performance (Appendix D.2).
- **W2:** We clarified the distinction between PEFT limitations and federated-specific constraints, highlighting that DevFT is an enabling necessity for edge devices.
- **W3:** We clarified the rationale for Deconfliction-Guided Layer Grouping (Section 3.2) and validated it with an ablation study (Table 2).
- **W4:** We resolved the perceived inconsistency by distinguishing between the macro-objective (capacity growth) and micro-optimization (fusion fidelity).
- **Q1&Q2:** We validated DevFT's robustness against both static Non-IID data (Section 4.6) and dynamic data distributions caused by evolving client pools (Appendix D.2), and clarified the scope of federated-specific literature in Section 2.1.
- **Q3&4:** We clarified the hierarchical structure of developmental stages versus communication rounds (Section 2.2) and explained how DBLF preserves intrinsic non-linear capabilities.

**DevFT Authors**

---

### Meta-Review · Area_Chair_ipgY · 2026-01-02

**Summary:**

This paper proposes Developmental Federated Tuning (DevFT), a resource-efficient federated fine-tuning paradigm for large language models (LLMs), inspired by cognitive development. The authors decompose the training process into progressive stages of increasing model capacity, enabling low-resource clients to contribute in early stages. Two key technical contributions—Deconfliction-Guided Layer Grouping (DGLG) and Differential-Based Layer Fusion (DBLF)—are used to construct stage-specific submodels.

While reviewers acknowledged the motivation, technical soundness, and comprehensive experimentation, several concerns were raised:

***Conceptual clarity and motivation***: Some reviewers (e.g., fj71, wUou) found the motivation unclear or inconsistent, particularly regarding why devices with insufficient resources for the full model would benefit from early-stage participation.\
***Relationship to existing methods***: Reviewers noted insufficient discussion of related resource-aware FL and PEFT methods (e.g., FedPilot, FedHeLLo, FlexLoRA).\
***Methodological clarity***: There were questions about the clarity of system-level design (e.g., what Eq. 5 does), the grouping/fusion mechanisms, and how convergence is achieved.\
***Theoretical analysis***: The convergence guarantee was seen as sketchy or incomplete by some, especially regarding the impact of layer fusion (4FbG, Sn7Y).\
***Baseline coverage and scalability***: Reviewers questioned the small number of clients used in experiments and whether the method scales to realistic FL settings (Sn7Y, 4FbG).\
***Applicability to non-IID and heterogeneous client settings***: Whether DevFT performs robustly under real-world FL heterogeneity was an open question.

**Final Recommendation: Accept (Poster)**\
The authors have substantially improved the paper during the rebuttal, addressing nearly all technical and empirical concerns with thorough new experiments, theoretical analysis, and clarifications. While one reviewer remains unconvinced on conceptual grounds, the overall contribution is novel, well-motivated, and empirically strong. The method offers a practical, extensible solution to enable edge participation in federated LLM fine-tuning, which is an important and timely problem.

**Reviewer Concerns:**

**Addressed Concerns**\
The authors made substantial efforts to address each reviewer's concerns with expanded experiments, clearer explanations, and new theoretical results. Key improvements include:

***Motivation and practicality***: The authors clarified that early-stage participation allows low-resource devices to contribute valuable data that is encoded into later stages, even if they drop out later. They demonstrated that this leads to significantly improved performance over methods that exclude such clients.\
***New baselines and scalability***: New experiments include comparisons with FedHeLLo, FlexLoRA, HETLoRA, and AFL/q-FFL under various compression and client settings. They also scale experiments to 10,000 clients with non-IID data, showing strong performance.\
***Theoretical analysis***: The authors included a formal convergence proof in Appendix A, accounting for the approximation bias introduced by DBLF.\
***Clarified system design***: The authors clarified the role of developmental stages, how Eq. 5 is used to construct submodels (not for aggregation), and how model updates work across stages.\
***Layer grouping and fusion***: DGLG and DBLF are explained in more detail, with ablation studies showing their empirical benefits and robustness to different fusion strategies.\
***Hyperparameter sensitivity***: New sensitivity analyses for LoRA rank, anchor layer choice, fusion weight β, initial capacity, and growth rate were added.

**Remaining or Partially Addressed Concerns**

***FL-specific positioning***: Some concerns remain about whether the method solves a problem specifically tied to FL or more generally to LLM fine-tuning. The authors argue persuasively that DevFT’s impact is strongest in FL, where resource constraints are most severe.

***Code release***: While the authors claim code is submitted with the paper, no explicit public link was confirmed.

**Reviewer Scores:**

**Reviewer fj71 (Initial Score: 4)**: Raised detailed concerns on motivation, system design, peak memory, aggregation, baselines, and theoretical analysis. Authors responded thoroughly with new experiments and clarifications. Score likely to increase to 6, assuming the reviewer accepted the clarifications and evidence provided.

**Reviewer 4FbG (Initial Score: 6)**: Initially positive and supported the method’s novelty and soundness but asked for more analysis on approximation bias, hyperparameters, and resource accounting. Authors fully addressed these in the revision. Score likely to remain at 6, or potentially further increase to 8.

**Reviewer Sn7Y (Initial Score: 4)**: Concerned about baselines, scalability, lack of convergence analysis, and the FL relevance of the method. The authors added large-scale experiments, new baselines, and clarified theoretical analysis. Score could reasonably increase to  6, though original review was skeptical.

**Reviewer wUou (Initial Score: 4)**: Focused on conceptual clarity and FL relevance. The authors provided strong rebuttals and empirical support. Score likely remains at 4, though the concerns are now more philosophical than technical.

---

### Decision · Program_Chairs · 2026-01-26

Accept (Poster)